# Accurate recognition of colorectal cancer with semi-supervised deep learning on pathological images

Gang Yu [1], Kai Sun [1], Chao Xu [2], Xing-Hua Shi [3], Chong Wu [4], Ting Xie [1], Run-Qi Meng [5], Xiang-He Meng [6], Kuan-Song Wang [7✉], Hong-Mei Xiao [6✉] & Hong-Wen Deng [6,8✉]

Machine-assisted pathological recognition has been focused on supervised learning (SL) that suffers from a significant annotation bottleneck. We propose a semi-supervised learning (SSL) method based on the mean teacher architecture using 13,111 whole slide images of colorectal cancer from 8803 subjects from 13 independent centers. SSL (~3150 labeled, ~40,950 unlabeled; ~6300 labeled, ~37,800 unlabeled patches) performs significantly better than the SL. No significant difference is found between SSL (~6300 labeled, ~37,800 unlabeled) and SL (~44,100 labeled) at patch-level diagnoses (area under the curve (AUC): 0.980 ± 0.014 vs. 0.987 ± 0.008, P value = 0.134) and patient-level diagnoses (AUC: 0.974 ± 0.013 vs. 0.980 ± 0.010, P value = 0.117), which is close to human pathologists (average AUC: 0.969). The evaluation on 15,000 lung and 294,912 lymph node images also confirm SSL can achieve similar performance as that of SL with massive annotations. SSL dramatically reduces the annotations, which has great potential to effectively build expert-level pathological artificial intelligence platforms in practice.

[1] Department of Biomedical Engineering, School of Basic Medical Science, Central South University, 410013 Changsha, Hunan, China. [2] Department of Biostatistics and Epidemiology, University of Oklahoma Health Sciences Center, Oklahoma City, OK 73104, USA. [3] Department of Computer & Information Sciences, College of Science and Technology, Temple University, Philadelphia, PA 19122, USA. [4] Department of Statistics, Florida State University, Tallahassee, FL 32306, USA. [5] Electronic Information Science and Technology, School of Physics and Electronics, Central South University, 410083 Changsha, Hunan, China. [6] Center for System Biology, Data Sciences and Reproductive Health, School of Basic Medical Science, Central South University, 410013 Changsha, Hunan, China. [7] Department of Pathology, Xiangya Hospital, School of Basic Medical Science, Central South University, 410078 Changsha, Hunan, China. [8] Deming Department of Medicine, Tulane Center of Biomedical Informatics and Genomics, Tulane University School of Medicine, New Orleans, LA 70112, USA. ✉email: 375527162@qq.com; hmxiao@csu.edu.cn; hdeng2@tulane.edu

Colorectal cancer (CRC) is the second most common cause of cancer death in Europe and America[1,2]. Pathological diagnosis is one of the most authoritative methods for diagnosing CRC[3,4], which requires a pathologist to visually examine digital full-scale whole slide images (WSI). The challenges stem from the complexity of WSI including large image sizes (>10,000 × 10,000 pixels), complex shapes, textures, and histological changes in nuclear staining[4]. Furthermore, there is a shortage of pathologists worldwide in stark contrast with the rapid accumulation of WSI data, and the daily workload of pathologists is intensive which could lead to unintended misdiagnose due to fatigue[5]. Hence, it is crucial to develop diagnosing strategies that are effective yet of low cost by leveraging recent artificial intelligence (AI) development.

Deep learning provides an exciting opportunity to support and accelerate pathological analysis[6], including lung[7,8], breast[9], lymph node[10], and skin cancers[11,12]. Progress has been made in applying deep learning to CRC including classification[13], tumor cell detection[14,15], and outcome prediction[16-18]. We have developed a recognition system for CRC using supervised learning (SL), which achieved one of the highest diagnosis accuracies in cancer diagnosis with AI[19]. However, our earlier method was built upon learning from 62,919 labeled patches from 842 subjects, which were carefully selected and extensively labeled by pathologists.

While SL with massive labeled data can achieve high diagnostic accuracy, the reality is that we often have only a small amount of labeled data and a much larger amount of unlabeled data in medical domains. Although unsupervised learning does not require any labeled data, its performance is still limited currently[20,21]. There are some other approaches for learning on the small amount of labeled data. For example, in transfer learning, the network is firstly trained in a big data set of source domain, and then trained in labeled medical images. However, the number of labeled images needed is still quite large[22,23]. The generative adversarial networks (GAN) can generate a large amount of data by learning the style from a limited data set[24,25]. These approaches may improve accuracy, but they only used limited labeled data sets, and large amounts of unlabeled data do appear in medical domains and clinical settings. Moreover, it would be difficult for GAN to simulate all possible features of the disease based on limited samples.

The semi-supervised learning (SSL), a method that leverages both labeled and unlabeled data is supposed to provide a low-cost alternative in terms of the requirement of the laborious and sometimes impractical sample labeling[26,27]. Although SSL can improve the accuracy of natural images, its performance on medical images is unclear. Recently, some studies were proposed to determine whether SSL based on a small amount of labeled data and a large amount of unlabeled data can improve medical image analysis[28-30], such as object detection[31], data augmentation[32], image segmentation[33,34]. However, only a very limited few studies have investigated if SSL can be applied to achieve satisfactory accuracy in pathological images[35], where on a small data set of 115 WSIs, an SSL method of CRC recognition can achieve the best accuracy of 0.938 only at 7180 patches of 50 WSIs from one data center, suggesting the potential of SSL for pathological diagnosis on patch-level.

However, to the best of our knowledge, the CRC recognition system of SSL has not been extensively validated on patient-level data set from multiple centers to assess the general utility of SSL. How to translate the patch-level prediction to WSI and patient-level diagnosis is not trivial. Because we and other groups have not been able to develop perfect patch-level models, the errors at patch-level may be easily magnified on WSI level diagnosis. For example, even though the imperfect patch-level model may yield reasonable prediction on positive (cancerous) WSIs, it also may yield high false-positive errors on the negative (non-caner) WSIs, because the false-positive errors at patch level will accumulate due to the testing of multiple patches in WSI. However, the patient-level diagnosis is required in the clinical applications of any AI system for cancer diagnosis.

To fill this gap, we used 13,111 WSIs collected from 8803 subjects from 13 independent centers to develop a CRC semi-supervised model. We evaluated SSL by comparing its performance with that of prevailing SL and also with that of professional pathologists. To confirm that SSL can achieve excellent performance on pathological images and further demonstrate our main point that a reliable medical AI can be built with a small amount of labeled data plus other available unlabeled data, we evaluated it on two other types of cancer (lung cancer and lymphoma). The main contributions of this study are summarized as follows:

(1) We evaluated different CRC recognition methods based on SSL and SL at the patch-level and patient-level respectively. This large-scale evaluation showed that accurate CRC recognition is feasible with a high degree of reliability even when the amount of labeled data is limited.

(2) We found that when ~6300 labeled patches (assuming a large number of unlabeled patches (e.g., ~37,800) available, which was often the case in practice) were used for SSL, there was no significant difference between SSL and SL (developed based on ~44,100 labeled patches) and pathologists. This finding holds for CRC recognition at both the patch level and the patient level.

(3) The extended experiment of lung cancer and lymphoma further confirmed the conclusion that when a small amount of labeled data plus a large amount of unlabeled data were used, SSL may perform similarly or even better than SL. Our study thus indicated that SSL would dramatically reduce the amount of labeled data required in practice, to greatly facilitate the development and application of AI in medical sciences.

## Results

The evaluations were performed on patch-level and patient-level diagnosis. For simplicity, we used SSL, SL to represent semi-supervised and supervised learning methods.

**SSL vs. SL CRC recognition at patch level.** The 62,919 patches from 842 WSIs in Dataset-PATT were used for patch-level training and testing (Table 1 and Fig. 1). The 30% of 842 WSIs (~18,819 patches) were used for the testing, and the remaining 70% of the WSIs (~44,100 patches) were used for the training (Table 2). Model-5%-SSL and Model-10%-SSL were trained on 5% (~3150) and 10% (~6300) labeled patches, respectively, where the remaining 65% (~40,950) and 60% (~37,800) patches were used, but their labels were ignored (as unlabeled patches). Model-5%-SL and Model-10%-SL were trained on the same labeled patches (5%, ~3150 and 10%, ~6300) only with Model-5%-SSL and Model-10%-SSL respectively, but the remained patches (~40,950, ~37,800) were not used. Model-70%-SL was trained on the ~44,100 labeled training patches.

The area under the curve (AUC) distribution on Dataset-PATT and Dataset-PAT were shown in Fig. 2. Model-5%-SSL was superior to Model-5%-SL (average AUC and standard deviation in Dataset-PATT: $0.906 \pm 0.064$ vs. $0.789 \pm 0.016$, $P$ value=0.017; Dataset-PAT: $0.948 \pm 0.041$ vs. $0.898 \pm 0.029$, $P$ values = 0.017; Both Dataset-PATT and Dataset-PAT: $0.927 \pm 0.058$ vs. $0.843 \pm 0.059$, $P$ value = 0.002, Wilcoxon-signed rank test). Model-10%-SSL was also significantly better than Model-10%-SL (AUC in Dataset-PATT: $0.990 \pm 0.009$ vs. $0.944 \pm 0.032$, $P$ value = 0.012; Dataset-PAT: $0.970 \pm 0.012$ vs. $0.908 \pm 0.024$, $P$ values = 0.012; both: $0.980 \pm 0.014$ vs. $0.926 \pm 0.034$, $P$ value = 0.0004).

**Table 1 Data sets used from multi-center data sources.**

| Data source | Dataset usage | Sample preparation | Population | Examination type Radical surgery/colonoscopy | CRC Subjects | CRC Slides | Non-CRC Subjects | Non-CRC Slides | Total Subjects | Total Slides |
|---|---|---|---|---|---|---|---|---|---|---|
| Xiangya Hospital (XH, Dataset-PATT) | PATT[a] | FFPE[b] | Changsha, China | 100%/0% | 614 | 614 | 228 | 228 | 842 | 842 |
| NCT-UMM (Dataset-PAT) | PAT[c] | FFPE | Germany | NA[d] | NA | NA | NA | NA | NA | 86 |
| Xiangya Hospital (XH-Dataset-PT) | PT[e] | FFPE | Changsha, China | 80%/20% | 3990 | 7871 | 1849 | 2132 | 5839 | 10,003 |
| Xiangya Hospital (XH-Dataset-HAC) | PT & HAC[f] | FFPE | Changsha, China | 89%/11% | 98 | 99 | 97 | 114 | 195 | 213 |
| Pingkuang Collaborative Hospital (PCH) | PT & HAC | FFPE | Jiangxi, China | 60%/40% | 50 | 50 | 46 | 46 | 96 | 96 |
| The Third Xiangya Hospital (TXH) | PT & HAC | FFPE | Changsha, China | 61%/39% | 48 | 70 | 48 | 65 | 96 | 135 |
| Hunan Provincial People's Hospital (HPH) | PT & HAC | FFPE | Changsha, China | 61%/39% | 49 | 50 | 49 | 49 | 98 | 99 |
| Adicon clinical laboratory (ACL) | PT & HAC | FFPE | Changsha, China | 22%/78% | 100 | 100 | 107 | 107 | 207 | 207 |
| Fudan University Shanghai Cancer Center (FUS) | PT & HAC | FFPE | Shanghai, China | 97%/3% | 100 | 100 | 98 | 98 | 198 | 198 |
| Guangdong Provincial People's Hospital (GPH) | PT & HAC | FFPE | Guangzhou, China | 77%/23% | 100 | 100 | 85 | 85 | 185 | 185 |
| Southwest Hospital (SWH) | PT & HAC | FFPE | Chongqing, China | 93%/7% | 99 | 99 | 100 | 100 | 199 | 199 |
| The First Affiliated Hospital of Air Force Medical University (AMU) | PT & HAC | FFPE | Xi'an, China | 95%/5% | 101 | 101 | 104 | 104 | 205 | 205 |
| Sun Yat-Sen University Cancer Center (SYU) | PT & HAC | FFPE | Guangzhou, China | 100%/0% | 91 | 91 | 6 | 6 | 97 | 97 |
| Chinese PLA General Hospital (CGH) | PT | FFPE | Beijing, China | 100%/0% | 0 | 0 | 100 | 100 | 100 | 100 |
| The Cancer Genome Atlas (TCGA-FFPE) | PT | FFPE | U.S. | 100%/0% | 441 | 441 | 5 | 5 | 446 | 446 |
| Total | | | | | 5881 | 9786 | 2922 | 3239 | 8803 | 13,111 |

a Patch-level training and test.
b Formalin-fixed and paraffin-embedded.
c Independent patch-level test.
d No information (NA) on the number of cancer or non-cancer subjects/slides were provided.
e Patient-level test.
f Human-AI competition.

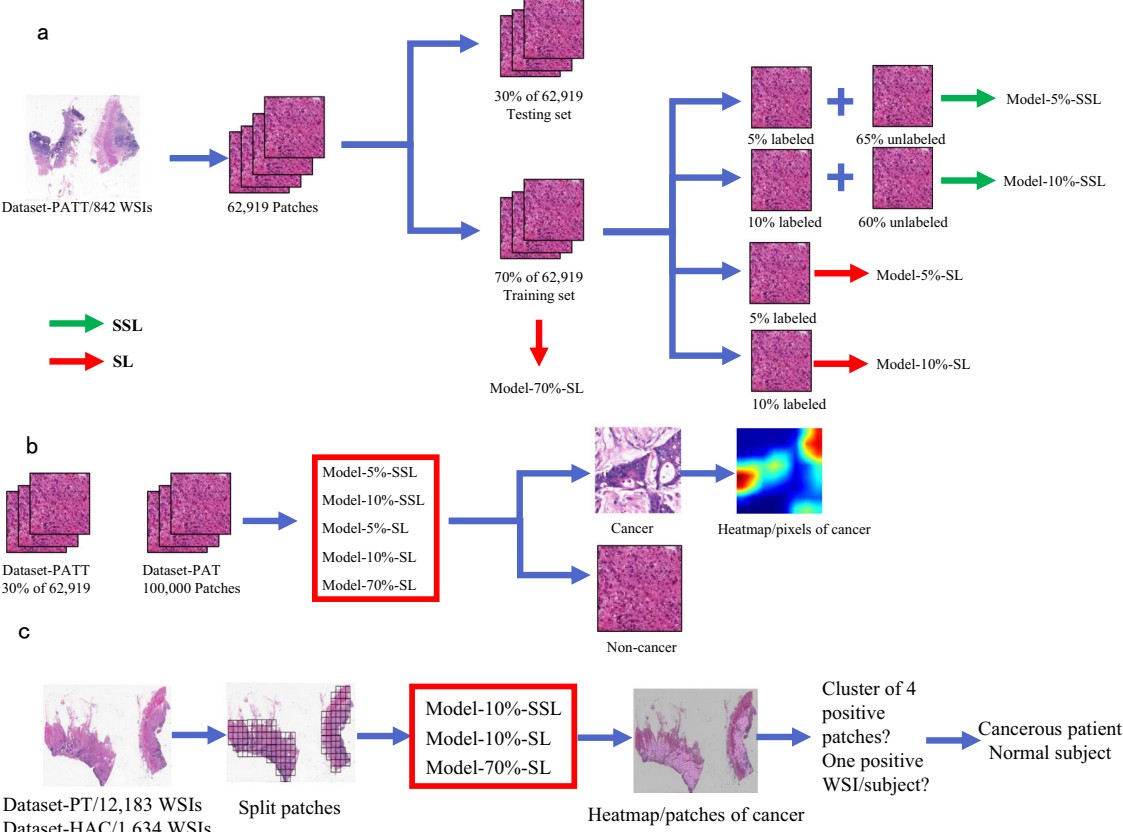

**Fig. 1 The flow chart of the colorectal cancer study. a** Semi-supervised learning (SSL) and supervised learning (SL) are performed on different labeled and unlabeled patches from 70% whole slide images (WSIs) of Dataset-PATT. Model-5%/10%-SSL and Model-5%/10%/70%-SL are obtained. **b** The patch-level test is performed on the patches from 30% WSIs of Dataset-PATT and whole data set of Dataset-PAT, and the above five models predict whether there is cancer or not in the patches. **c** The patient-level test and human-AI competition are performed on Dataset-PT and Dataset-HAC, respectively. Each WSI is divided into many patches, and three models infer whether these patches are cancerous or normal individually. The clustering-based method is then used on the WSI. If there is a cluster of four positive patches on a WSI, the WSI is positive. A subject with one or more positive WSIs is cancerous, or the subject is normal.

**Table 2 Training and testing set for CRC patch-level models.**

| Model | Class | Dataset-PATT | | | Dataset-PAT |
|---|---|---|---|---|---|
| | | Training set | | Testing set | |
| | | Labeled | Unlabeled[a] | | |
| Model-5%-SSL | Cancer | 1645 | 21,390 | 9828 | 14,317 |
| | Non-cancer | 1505 | 19,560 | 8991 | 85,683 |
| | Total | 3150/5%[b] | 40,950/65%[c] | 18,819/30%[d] | 100,000 |
| Model-10%-SSL | Cancer | 3290 | 19,745 | 9828 | 14,317 |
| | Non-cancer | 3010 | 18,055 | 8991 | 85,683 |
| | Total | 6300/10% | 37,800/60%[e] | 18,819/30% | 100,000 |
| Model-5%-SL | Cancer | 1645 | – | 9828 | 14,317 |
| | Non-cancer | 1505 | – | 8991 | 85,683 |
| | Total | 3150/5% | – | 18,819/30% | 100,000 |
| Model-10%-SL | Cancer | 3290 | – | 9828 | 14,317 |
| | Non-cancer | 3010 | – | 8991 | 85,683 |
| | Total | 6300/10% | – | 18,819/30% | 100,000 |
| Model-70%-SL | Cancer | 23,035 | – | 9828 | 14,317 |
| | Non-cancer | 21,065 | – | 8991 | 85,683 |
| | Total | 44,100/70%[f] | – | 18,819/30% | 100,000 |

[a]The labels of the patches are ignored.
[b–f]Because the number of patches from each WSI is not the same, the number of patches estimated based on the proportion of extraction is approximate.

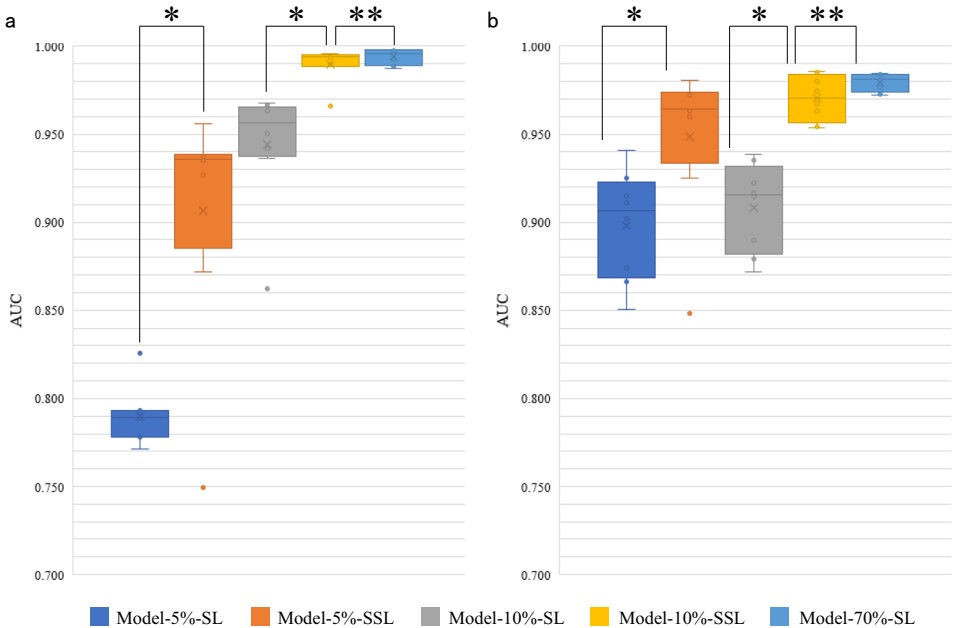

**Fig. 2 Area under the curve (AUC) distribution of five models at patch level.** The boxes indicate the upper and lower quartile values, and the whiskers indicate the minima and maxima values. The horizontal bar in the box indicates the median, while the cross indicates the mean. The circles represent data points, and the scatter dots indicate outliers. * indicates significant difference, and ** indicates no significant difference. **a** The evaluation of five models on the testing set of Dataset-PATT. Eight versions of each model (number of experiments per model = 8) are tested at their testing sets (number of samples/ patches per testing set = ~18,819) independent from their training sets, respectively. The Wilcoxon-signed rank test is then used to evaluate the significant difference of AUCs (sample size/group = 8) between two models. Two-sided $P$ values are reported, and no adjustment is made. The average AUC and standard deviation of Model-5%-SSL and Model-5%-SL: 0.906 ± 0.064 vs. 0.789 ± 0.016, $P$ value = 0.017; Model-10%-SSL and Model-10%-SL: 0.990 ± 0.009 vs. 0.944 ± 0.032, $P$ value = 0.012; Model-10%-SSL and Model-70%-SL: 0.990 ± 0.009 vs. 0.994 ± 0.004, $P$ value = 0.327. **b** The evaluation of 8 versions of five models on the Dataset-PAT (number of samples/patches per testing set = 100,000). Wilcoxon-signed rank test (sample size/group = 8), and two-sided $P$ values are reported. Model-5%-SSL and Model-5%-SL: 0.948 ± 0.041 vs. 0.898 ± 0.029, $P$ values = 0.017; Model-10%-SSL and Model-10%-SL: 0.970 ± 0.012 vs. 0.908 ± 0.024, $P$ value = 0.012; Model-10%-SSL and Model-70%-SL: 0.970 ± 0.012 vs. 0.979 ± 0.005, $P$ values = 0.263. The AUC values of each model on Dataset-PATT and Dataset-PAT are combined and the Wilcoxon-signed rank test is performed on the combined results (sample size/group = 16), and two-sided $P$ values are reported. Model-5%-SSL and Model-5%-SL: 0.927 ± 0.058 vs. 0.843 ± 0.059, $P$ value = 0.002; Model-10%-SSL and Model-10%-SL: 0.980 ± 0.014 vs. 0.926 ± 0.034, $P$ value = 0.0004; Model-10%-SSL and Model-70%-SL: 0.980 ± 0.014 vs. 0.987 ± 0.008, $P$ value = 0.134.

These results indicated that when ~3150 (5%) or 6300 (10%) patches were labeled, the SSL was always better than SL.

The performance of Model-10%-SSL had no significant difference with that of Model-70%-SL (AUC in Dataset-PATT: 0.990 ± 0.009 vs. 0.994 ± 0.004, $P$ value = 0.327; Dataset-PAT: 0.970 ± 0.012 vs. 0.979 ± 0.005, $P$ values = 0.263; both: 0.980 ± 0.014 vs. 0.987 ± 0.008, $P$ value = 0.134). This observation indicated that there was no significant difference between the SSL (6300 labeled, 37,800 unlabeled) and the SL (44,100 labeled). Visual inspection (Supplementary Fig. 2) confirmed that that Model-10%-SL could not really find the locations of cancer in the patches, while the locations of cancer by Model-10%-SSL and Model-70%-SL were highly matched.

**Patient-level CRC recognition.** To test whether the above conclusion at patch-level still holds at patient-level, we evaluated three of five models using Dataset-PT. The patient-level diagnosis was based on the recognition of every patch provided by patch-level models, and then cluster-based WSI inference and positive sensitivity for patient inference (Fig. 1c). The results were shown in Fig. 3.

Model-10%-SSL had a significant improvement over Model-10%-SL (AUC: 0.974 ± 0.013 vs. 0.819 ± 0.104, $P$ value = 0.002) on patient-level prediction in the multi-centers scenario. The AUC of Model-10%-SSL was slightly lower than, but comparable to, that of Model-70%-SL (AUC: 0.974 ± 0.013 vs. 0.980 ± 0.010, $P$

value = 0.117). Among the 7 data sets (XH-Dataset-PT, XH-Dataset-HAC, PCH, TXH, FUS, SWH, TCGA-FFPE, 11,290 WSIs), the AUC difference of Model-10%-SSL and Model-70%-SL was smaller than 0.016. In particular, on the largest data set, XH-dataset-PT (10,003 WSIs), the AUCs of Model-10%-SSL and Model-70%-SL were close with 0.984 vs. 0.992. On the data sets of HPH, SYU, CGH, and AMU (501 WSIs), the AUCs of Model-10%-SSL were even higher than that of Model-70%-SL.

In the data from GPH, and ACL (392 WSIs), the performance of Model-10%-SSL was lower than that of Model-70%-SL (AUC DIFF ≥ −0.040). It was worth noting that Model-10%-SSL generally achieved good sensitivity, which proved practically useful for the diagnosis of CRC. Visual inspection in Supplementary Fig. 3 showed the cancer patches identified by Model-10%-SSL and Model-70%-SL were the true cancer locations on WSIs.

**Human-AI CRC competition.** We recruited six pathologists with 1–18 years of independent experience (Supplementary Table 1). They independently reviewed 1634 WSIs/1576 subjects from 10 data centers (Supplementary Table 3, Dataset-HAC, Fig. 4) with no time limit and diagnosed cancer solely based on WSIs (i.e., no other clinical data were used). We ranked pathologists, Model-10%-SSL and Model-70%-SL. The average AUC of model-10%-SSL was 0.972, ranked at the 5th, which was close to the average AUC of pathologists (0.969).

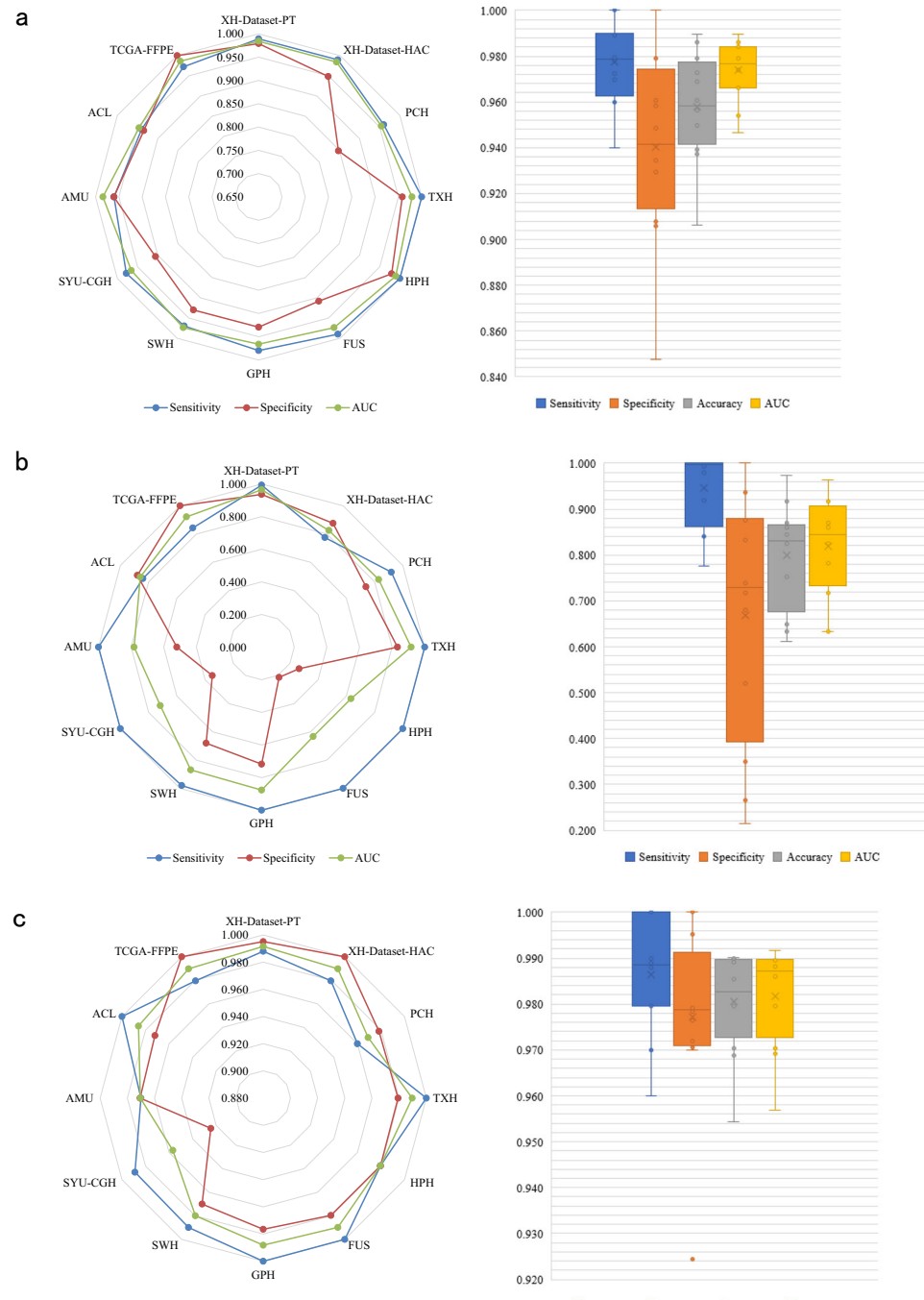

**Fig. 3 The results of patient-level CRC recognition.** Patient-level comparison of **a** Model-10%-SSL, **b** Model-10%-SL, and **c** Model-70%-SL on 12 independent data sets from Dataset-PT. Left: Radar maps illustrate the sensitivity, specificity, and area under the curve (AUC) of three models on 12 centers. Right: Boxplots show the distribution of sensitivity, specificity, accuracy, and AUC of the three models in these centers. The boxes indicate the upper and lower quartile values, and the whiskers indicate the minima and maxima values. The horizontal bar in the box indicates the median, while the cross indicates the mean. The circles represent data points, and the scatter dots indicate outliers. The average AUC and standard deviation (sample size = 12) are calculated for each model, and the Wilcoxon-signed rank test (sample size/group = 12) is then used to evaluate the significant difference of AUCs between two models. Two-sided $P$ values are reported, and no adjustment is made. Model-10%-SSL vs. Model-10%-SL: AUC: 0.974 ± 0.013 vs. 0.819 ± 0.104, $P$ value = 0.002; Model-10%-SSL vs. Model-70%-SL: 0.974 ± 0.013 vs. 0.980 ± 0.010, $P$ value = 0.117. The data points are listed in Supplementary Data 1.

**Extended SSL vs. SL experiment of lung and lymph node cancer.** In order to demonstrate the utility of SSL on other pathological images, the experiments of SSL and SL on lung and lymph node were performed. 15,000 lung images of three classes: adenocarcinoma, squamous cell carcinoma, and benign tissue were obtained from LC25000 dataset (Lung)[36], and the

294,912 lymph node images including tumor and benign tissue were obtained from PatchCamelyon dataset (Pcam)[37]. Similarly, SSL was trained on a small number of labeled images and a large number of unlabeled images (for which labels are known but ignored during training), and compared with SL (Tables 3 and 4).

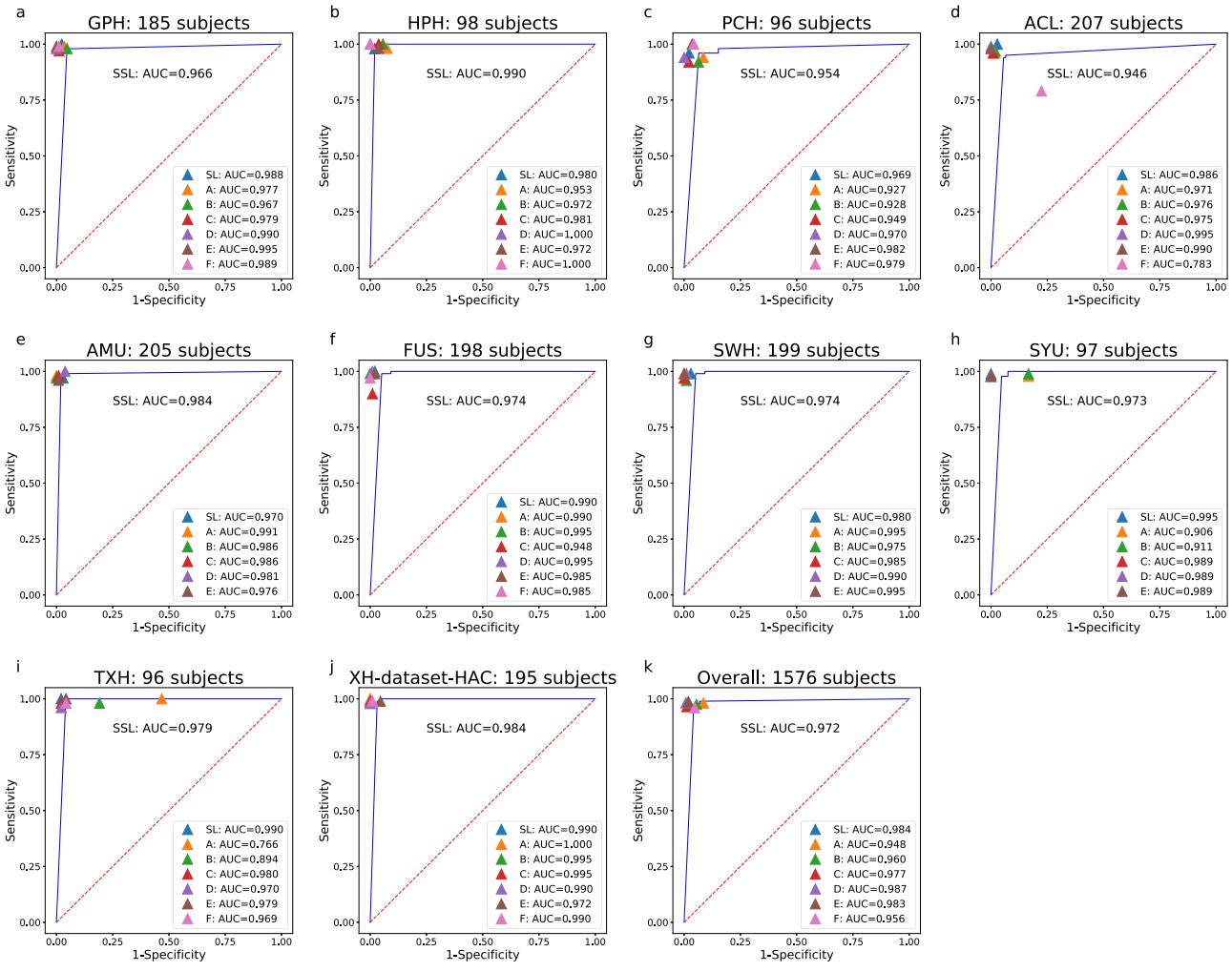

**Fig. 4 The Human-AI CRC competition results.** Area under the curve (AUC) comparison of Model-10%-SSL(SSL), Model-70%-SL(SL) and six pathologists (**a–f**) using Dataset-HAC, which consists XH-dataset-HAC, PCH, TXH, HPH, ACL, FUS, GPH, SWH, AMU and SYU. Blue lines indicate the AUCs achieved by Model-10%-SSL. The F pathologist did not attend the competition of SYU, AMU, and SWH data set.

Because the number of classes in lung images was three and the number of images in each class was balanced (5000 per class), the accuracy was used for the evaluation. Lung-5%-SSL (5% labeled and 75% unlabeled) and Lung-20%-SSL (20% labeled, 60% unlabeled) were better than Lung-5%-SL (5% labeled) and Lung-20%-SL (20% labeled) (average accuracy and standard deviation: $0.960 \pm 0.006$ vs. $0.918 \pm 0.023$, $P$ value $= 0.012$; $0.989 \pm 0.003$ vs. $0.961 \pm 0.022$, $P$ value $= 0.011$, Fig. 5), respectively. There was no difference between Lung-20%-SSL and Lung-80%-SL (80% labeled) (accuracy: $0.989 \pm 0.003$ vs. $0.993 \pm 0.002$, $P$ value $= 0.093$). Pcam-1%-SSL (1% labeled, 99% unlabeled) and Pcam-5%-SSL (5% labeled, 95% unlabeled) were better than Pcam-1%-SL (1% labeled) and Pcam-5%-SL (5% labeled) (average AUC and standard deviation: $0.947 \pm 0.008$ vs. $0.912 \pm 0.008$, $P$ value $= 0.012$; $0.960 \pm 0.002$ vs. $0.943 \pm 0.009$, $P$ value $= 0.011$, Fig. 6), respectively. Pcam-5%-SSL can be compared to Pcam-100%-SL (100% labeled) (AUC: $0.960 \pm 0.002$ vs. $0.961 \pm 0.004$, $P$ value $= 0.888$). This extended experiment confirmed the conclusion that when a small number of labeled pathological images were available together with a large number of unlabeled image data, SSL can be compared to SL with massive labels.

**Comparison with related research**. We compared our methods with seven existing CRC detection methods[13,17,35,38–41], and

five other cancers (lung, ductal carcinoma, breast, prostate, basal cell carcinoma) detection methods[7,42–45] (Supplementary Table 2). The 6 of 7 CRC detection methods had an AUC ranging from 0.904 to 0.99 based on SL. Besides, Shaw et al.[35] used cancer and normal patches in 86 CRC WSIs to develop an SSL detection method, and used the test set of 7180 patches in 50 WSIs with colorectal adenocarcinoma, all from one data center, with the best accuracy of 0.938 confirming the potential of SSL on patch-level. In this study, we showed the advantages of the SSL method with 162,919 patches and 13,111 WSIs at both patch and patient levels from multiple independent centers, attesting to the robustness and general utility of the SSL model we developed, where the Model-10%-SSL was comparable to the recent SL model[19]. Besides, Lung-20%-SSL was also comparable to the SL of Coudray et al. for lung cancer detection[7].

## Discussion
Accurately diagnosing CRC requires years of training, leading to a global shortage of pathologists[2]. Almost all existing computer-assisted diagnosis models currently rely on massive labeled data with SL, but manual labeling is usually time-consuming and costly. This leads to an increasing interest in building an accurate diagnosis system with far less labeled data.

**Table 3 Training and testing set for lung models.**

| Model | Class | Dataset-lung Training set | | Testing set |
|---|---|---|---|---|
| | | Labeled | Unlabeled | |
| Lung-5%-SSL | Adenocarcinoma | 250 | 3750 | 1000 |
| | Squamous cell carcinoma | 250 | 3750 | 1000 |
| | Benign | 250 | 3750 | 1000 |
| | Total | 750/5% | 11,250/75% | 3000/20% |
| Lung-20%-SSL | Adenocarcinoma | 1000 | 3000 | 1000 |
| | Squamous cell carcinoma | 1000 | 3000 | 1000 |
| | Benign | 1000 | 3000 | 1000 |
| | Total | 3000/20% | 9000/60% | 3000/20% |
| Lung-5%-SL | Adenocarcinoma | 250 | – | 1000 |
| | Squamous cell carcinoma | 250 | – | 1000 |
| | Benign | 250 | – | 1000 |
| | Total | 750/5% | – | 3000/20% |
| Lung-20%-SL | Adenocarcinoma | 1000 | – | 1000 |
| | Squamous cell carcinoma | 1000 | – | 1000 |
| | Benign | 1000 | – | 1000 |
| | Total | 3000/20% | – | 3000/20% |
| Lung-80%-SL | Adenocarcinoma | 4000 | – | 1000 |
| | Squamous cell carcinoma | 4000 | – | 1000 |
| | Benign | 4000 | – | 1000 |
| | Total | 12,000/80% | – | 3000/20% |

**Table 4 Training and testing set for lymph node models.**

| Model | Class | Dataset-Pcam Training set | | Testing set |
|---|---|---|---|---|
| | | Labeled | Unlabeled | |
| Pcam-1%-SSL | Tumor | 1311 | 129,761 | 16,384 |
| | Non-tumor | 1311 | 129,761 | 16,384 |
| | Total | 2622/1%[a] | 259,522/99%[b] | 32,768 |
| Pcam-5%-SSL | Tumor | 6554 | 124,518 | 16,384 |
| | Non-tumor | 6554 | 124,518 | 16,384 |
| | Total | 13,108/5%[c] | 249,036/95%[d] | 32,768 |
| Pcam-1%-SL | Tumor | 1311 | – | 16,384 |
| | Non-tumor | 1311 | – | 16,384 |
| | Total | 2622/1% | – | 32,768 |
| Pcam-5%-SL | Tumor | 6554 | – | 16,384 |
| | Non-tumor | 6554 | – | 16,384 |
| | Total | 13,108/5% | – | 32,768 |
| Pcam-100%-SL | Tumor | 131,072 | – | 16,384 |
| | Non-tumor | 131,072 | – | 16,384 |
| | Total | 262,144/100%[e] | – | 32,768 |

[a–e]1%, 99%, 5%, 95%, 100% of training set.

Although studies have shown that SSL achieved good results in tasks like natural image processing, SSL has not been widely evaluated for analyzing pathological images. In this study, we applied SSL to CRC diagnosis, and evaluated its performance using an extensive collection of WSIs across 13 medical centers. On this large data set, we conducted a range of comparison of CRC recognition performance among SSL, SL, and six human pathologists, at both patch level and patient level.

We demonstrated that SSL outperformed SL at patch-level recognition when only a small amount of labeled and large amounts of unlabeled data were available. In our previous study[19], we used 62,919 labeled patches from 842 WSIs, which achieved accurate patch-level recognition. When SSL was used as demonstrated in this study, only about a tenth (6300) of those many labeled patches plus 37,800 unlabeled patches were used to achieve similar AUC.

We also conducted extensive testing of three models for patient-level prediction on 12 centers (Dataset-PT). Just like the patch level, at the patient level, the SSL outperformed the SL when a small number of labeled patches were available, and close to SL when using a large number of labeled patches. The AUC of Model-10%-SL at XH-Dataset-PT was 0.964, perhaps because both the testing data and training data were from XH.

However, using the data from 12 centers, the average AUC of Model-10%-SL was dramatically reduced to 0.819 from 0.964. This result showed that when training data and testing data were not the same source, the generalization performance of Model-10%-SL was significantly reduced. Moreover, many cancerous patches predicted by Model-10%-SL were deviated from true cancer locations in a WSI (Supplementary Fig. 3).

When a large number of unlabeled patches were added for SSL, the generalization performance across centers can be maintained,

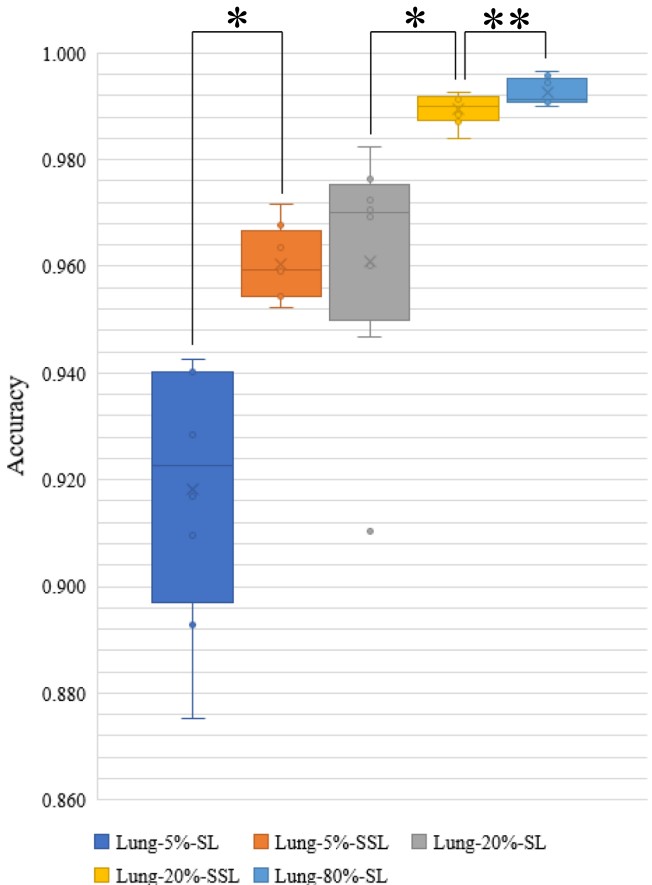

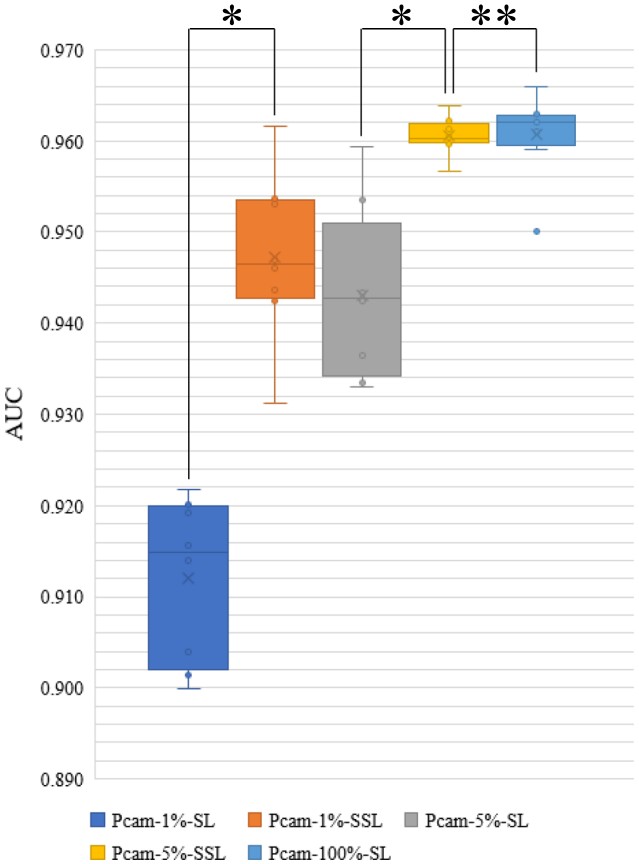

**Fig. 5 Accuracy distribution of five models on the testing set of LC25000 dataset (number of samples/patches per testing set = 3000; number of experiments per model = 8).** The boxes indicate the upper and lower quartile values, and the whiskers indicate the minima and maxima values. The horizontal bar in the box indicates the median, while the cross indicates the mean. The circles represent data points, and the scatter dots indicate outliers. * indicates significant difference, and ** indicates no significant difference. The Wilcoxon-signed rank test (sample size/group = 8) is then used to evaluate the significant difference in the accuracy between two models. Two-sided *P* values are reported, and no adjustment is made. The average AUC and standard deviation (sample size = 8) are calculated for each model. Lung-5%-SSL vs. Lung-5%-SL: 0.960 ± 0.006 vs. 0.918 ± 0.023, *P* value = 0.012; Lung-20%-SSL vs. Lung-20%-SL: 0.989 ± 0.003 vs. 0.961 ± 0.022, *P* value = 0.011; Lung-20%-SSL vs. Lung-80%-SL: 0.989 ± 0.003 vs. 0.993 ± 0.002, *P* value = 0.093.

**Fig. 6 Area under the curve (AUC) distribution of five models on the testing set of PatchCamelyon data set (number of samples/patches per testing set = 32,768; number of experiments per model = 8).** The boxes indicate the upper and lower quartile values, and the whiskers indicate the minima and maxima values. The horizontal bar in the box indicates the median, while the cross indicates the mean. The circles represent data points, and the scatter dots indicate outliers. * indicates significant difference, and ** indicates no significant difference. The Wilcoxon-signed rank test (sample size/group = 8) is then used to evaluate the significant difference of AUCs between the two models. Two-sided *P* values are reported, and no adjustment is made. The average AUC and standard deviation (sample size = 8) are calculated for each model. Pcam-1%-SSL vs. Pcam-1%-SL: 0.947 ± 0.008 vs. 0.912 ± 0.008, *P* value = 0.012; Pcam-5%-SSL vs. Pcam-5%-SL: 0.960 ± 0.002 vs. 0.943 ± 0.009, *P* value = 0.011; Pcam-5%-SSL vs. Pcam-100%-SL: 0.960 ± 0.002 vs. 0.961 ± 0.004, *P* value = 0.888.

where there was no significant difference when comparing with the accurate SL model using massive labeled patches[19]. These results showed that when labeled patches were seriously insufficient, using unlabeled data can greatly improve the generalization ability across different data sets. The patient-level results indicated that with SSL, we may not need as much labeled data as in SL. Since it is well known that unlabeled medical data are relatively easy to obtain, it is of great importance and with an urgent need to develop SSL methods.

We compared the diagnosis of six pathologists with SSL. We found that SSL reached an average AUC of pathologists, which was approximately equivalent to a pathologist with five years of clinical experience. The Human-AI competition in this regard thus showed that it was feasible to build an expert-level method for clinical practice based on SSL.

Based on the extended experiments of cancers of lung and lymph node, we further confirmed the conclusion on CRC that

when a small amount of labeled data was used, SSL plus a large amount of unlabeled data performed better than SL (with the same number of labeled images). SSL performance can be compared to SL with massive annotations, which confirms the conclusion that SSL may reduce the need for the amount of annotation data on pathological images.

In practice, the exact amount of the data that needs to be labeled is generally unknown. Nonetheless, as shown in our experiments, it is an alternative low-cost approach to conduct SSL. Hence, it is an effective strategy to wisely utilize all data so that a small amount of data is first labeled to build a baseline model based on SSL. If the results are not satisfactory for this baseline model, the amount of labeled data should be increased. This strategy is feasible since as expected, SSL requires a much smaller amount of labeled data to achieve the same performance compared with SL.

Our work confirmed that unlabeled data could improve the accuracy on insufficient labeled pathological images. We

 9

demonstrated that SSL with a small amount of labeled data of three cancers achieved comparable prediction accuracy as that of SL with massive labeled data and that of experienced pathologists. SSL may have excellent potentials to overcome the bottleneck of insufficient labeled data as in many medical domains. This study thus supported the potential applications of SSL to develop medical AI systems. In addition, we have noticed some other recent works[46,47], which have made a new strategy on the sparse and incomplete annotations to reduce the annotation effort for cell detection. This strategy is also applicable to annotations in our WSIs, and the unlabeled data is useful for SSL. In future work, how to make annotations and use unlabeled data more effectively should be further studied to improve the efficiency of medical AI development.

## Methods

**Ethics**. This study was approved by the Institutional Review Board of Xiangya School of Medicine, Central South University. Due to the retrospective nature of the study, informed consent was waived for the de-identified medical record data used in this study.

**Data sets**. Our CRC dataset was composed of 13,111 WSIs collected from 13 sources, including 10 hospitals, a professional adicon clinical laboratory (ACL) and two public databases (Table 1). The CRC WSIs were then divided into four data sets for different aims (Dataset-PATT, Dataset-PAT, Dataset-PT, Dataset-HAC, Supplementary Table 3). All WSIs were made from formalin-fixed and paraffin-embedded (FFPE) method.

Dataset-PATT was used for patch-level training and testing, Dataset-PAT for independent patch-level test. All the images from other hospitals as well as ACL (Dataset-PT) were used for patient-level testing. Dataset-HAC was used for human-AI competition.

Dataset-PATT included 62,919 patches (cancer 30,056, non-cancer 32,863) from 842 subjects (cancer 614, non-cancer 228, Table 1) from Xiangya Hospital (XH). The Dataset-PAT (NCT-CRC-HE-100K) from NCT biobank and the UMM pathology archive (NCT-UMM, National Center for Tumor diseases, University Medical Center Mannheim, Heidelberg University, Germany) was used for further patch-level validation, where there were 100,000 patches from 86 slides of CRC tissue. All the patches can be downloaded at https://zenodo.org/record/1214456#.XV2cJeg3lhF, whose labels were from the NCT-UMM website.

The Dataset-PT consisted of 12,183 WSIs from 10 hospitals, ACL and the cancer genome atlas (TCGA-FFPE, https://portal.gdc.cancer.gov/), which were used for extensive patient-level prediction. The WSIs from 9 of 13 centers and 213 WSIs from XH-Dataset-HAC were included to Dataset-HAC for human-AI competition after checking their labels carefully. Because XH was the biggest data source, the WSIs from XH were distributed independently and exclusively in Dataset-PATT, XH-Dataset-PT, and XH-Dataset-HAC.

**Digitization and annotation of pathological slides**. In the 10 hospitals and ACL, the technicians randomly selected slides from the archive library. The slides from 2010 to 2019 were scanned with a KF-PRO-005 scanner (KFBIO company, Ningbo City, China) at ×20 magnification. The number of selected patients collected on the same day was limited to less than 50 to make sure the selected WSIs for this study were not unduly influenced by samples collected on any one single day.

All diagnosis of images from TCGA, NCT-UMM were available online, and their labels were used directly. The WSIs from the 10 hospitals and ACL in Dataset-PT were independently reviewed by two senior and seasoned pathologists. When their diagnoses were consistent, the WSI was then included. Dataset-HAC was used for human-AI competition, and the review criteria were more rigorous. The label in Dataset-HAC was more strictly checked by three senior highly experienced pathologists who independently reviewed the pathological images without knowing the previous clinical diagnosis. If a consensus was reached, the WSI was included; otherwise, two other independent pathologists would join the review. After a discussion among the five pathologists, the WSI was included for the Human-AI competition only if they reached an agreement.

**Annotation of patches in Dataset-PATT**. The presented approach was based on the patch-level prediction. There was high phenotypic diversity within tumor and among tumors, the representation of cancer tissue in patches seriously affects the training. Therefore, the patches in Dataset-PATT were carefully selected to include all common tumor histological subtypes, ensuring the selected patches were widely representative for practical diagnosis.

The technician randomly selected 842 slides from pathological archive library of Xiangya hospital and then scanned them using a KF-PRO-005 scanner (KFBIO company, Ningbo City, China) at ×20 magnification. Because the shape of the CRC tissue was more diverse than that of non-cancerous tissue, more cancer positive WSIs (614) and less cancer negative WSIs (228) were selected. For the 614 positive

WSIs, the numbers of positive WSIs of various CRC subtypes were basically consistent with the subtype morbidity in the population.

Two pathologists used image browser software provided by KFRIO company (Ningbo City, China) from one WSI to export some non-overlapping regions of interest (ROI) according to the size of WSI. In order to maintain the diversity of cancer cell distribution, the 4–10 positive ROIs were extracted from each positive WSI. In order to ensure that the number of positive ROIs and negative ROIs was balanced, the 10–25 ROIs were extracted from each negative WSI. One ROI had a size of about 1024 × 768 pixels, and was split into about 6 non-overlapping patches with 300 × 300 pixels in order to be adaptable to meet the input size of most neural networks. The two pathologists then manually reviewed the patches, each of which was weakly labeled with either cancer or cancer-free. When two pathologists reached a consensus on the annotation of patches, which were kept in the Dataset-PATT.

In total, 62,919 patches were obtained. The 30,056 labeled tumor patches from 614 patients and 32,863 normal patches from 228 healthy subjects were included in Dataset-PATT, that is, an average of 49 patches per cancerous WSI and 144 patches per healthy WSI were included. Meanwhile, the numbers of patches containing various proportions of cancer cells were approximately equal.

**Patch-level SSL and SL models**. The Dataset-PATT was randomly divided into training set and testing set according to the proportions shown in Table 2, and the patches from the same subject/WSI would not be in different sets, to ensure independence of the different data sets. Meanwhile, the patches from 70% of 842 subjects/WSIs were used as the training set, while the remaining patches from 30% subjects/WSIs were used as the testing set.

When the number of WSIs (70% of 842 WSIs) in the training set is known, there are two ways to reduce the labeling effort on the patches from these WSIs. The first method is that the patches from some WSIs are labeled, while the patches from other WSIs are unlabeled. However, there are some differences between WSIs such as staining, disease subtypes. SSL theoretically assumes that data points, both labeled and unlabeled, are smooth[48]. In other words, the labels of unlabeled patches are potentially determined by neighboring labeled patches in the feature space. If the labeled patches and the unlabeled patches come from different WSIs, the distance of labeled and unlabeled patch will unavoidably include the differences of colors and tissue structures among WSIs included in the training sets, thereby the smoothness assumption among data points is violated.

By contrast, because the labeled patches and unlabeled patches from the same WSI are similar and will not be affected by differences between WSIs. The smoothness assumption of SSL can be better met. Therefore, in order to extract n% (5%, 10%, or 70%) of total patches (62,919) as the labeled patches for training, we used another method that the n%/70% of all the patches from each WSI in the training set were randomly selected and labeled, and the remaining patches of the WSI were not labeled (labels were masked).

Five patch-level models (two SSL, three SL) were trained using labels of different portions of these patches (Table 2). In the training of Model-5%-SSL and Model-10%-SSL, we used SSL and kept labels for small proportions (i.e., 5% and 10%) of the total patches (62,919) and masked label information for the remaining patches (65% and 60%). In the training of Model-5%-SL, Model-10%-SL, and Model-70%-SL, we used SL with 5%, 10%, 70% of the total 62,919 patches.

**Algorithm pipeline**. Because WSI was very large (>10,000 × 10,000 pixels), the patches in a WSI were firstly extracted, and the patch-level models were trained to derive cancerous probability at patch-level. Finally, all the patch-level results on a WSI were combined to infer the cancerous probability of the WSI/patient. The flow chart is shown in Fig. 1.

**Patch-level SSL and SL**. The patch-level models included SL and SSL versions. For SL, the patches from the WSIs were input to the convolutional neural network (CNN). Our previous work tested some known CNNs, such as VGG[49], ResNet[50], Inception[51], and found that Inception V3 achieved the most consistent results on the CRC datasets[19]. Therefore, we used Inception V3 as the baseline model of SL. The patch size we labeled was 300 × 300, so we used the bilinear interpolation method[52] to scale the patch size to 299 × 299, which was the default input size of Inception V3. The top output layer was removed, and the output category was modified to two (cancer or non-cancer).

The SSL version was implemented based on the mean teacher method[26], where two Inception V3 were trained, one as a student and the other as teacher, which was one of SSL method (Supplementary Fig. 1). The student network used SL and required inputted patches, which included a small number of patches with labels and a large number of unlabeled patches. For the labeled patches, the cross-entropy of the predicted and real label was calculated as the classification cost. For unlabeled patches, the teacher network provided the pseudo labels, and the mean square error of the predicted labels and pseudo labels was calculated as the consistency cost. The sum of consistency cost and classification cost, as the total cost, was used for the student network training. In this study, the two networks were performed on the same architecture with SL, i.e., Inception V3.

**Network training at patch level**. The Inception V3 was initialized with the pre-trained model on ImageNet database[53], and then trained on the pathological images. During training, the weights in all layers of inception V3 were updated. We used the same preprocessing in protocols we used earlier[19]. All background patches without any cell tissue were removed. After data augmentation (image zoom, flip, color change), the grayscale of each pixel was normalized to [−1,1].

For each model, we adopted a general strategy where one-tenth of the labeled training set was taken out as the validation set for hyperparameters selection. The optimal hyperparameters with the highest accuracy in the validation set were selected for training the models. The parameters were listed in Supplementary Table 4.

In the SSL, because of the imbalance between the labeled and unlabeled data, we maintained the same proportion of labeled and unlabeled patches in each mini-batch of 128 patches. The optimizer was Adam. The training period was 500 epochs, and each epoch included 100 steps. If the accuracy on the validation set cannot be improved for 80 consecutive epochs, the early stopping[54] was applied. In order to prevent the training from ending prematurely, 50 epochs for pre-training were executed before the early stopping. L2 decay was used and the decay coefficient was set to 0.0001. The teacher network was initialized with the student network. The student network would update the weights in each step, but the teacher network used exponential moving average to update the weights after one epoch ended. The smoothing coefficient was set to 0.95.

In SL, the learning rate was 0.001, and the exponentially decay was used with a decay rate of 0.99. The number of epochs was 500, the steps per epoch were 100. The early stopping with patience 50 was also applied. The coefficient of L2 decay was 0.0001, and the batch size was 64.

**Clustered-based WSI inference**. Because the accuracy of patch-level models cannot be 100%, there were serious false positives in WSI predictions if any patch in the WSI was identified as positive (cancer) and used as a criterion for predicting the WSI cancerous status. Intuitively, because the tissues in WSI were continuous, the area with cancer should be distributed continuously and included several continuous positive patches. This intuition had been used to effectively control the false-positive of functional magnetic resonance images[55]. We designed a simple clustering-based inference method. If some continuous patches were identified as having cancer by the patch-level model, the cancer may indeed exist on WSI. For statistical analysis on patient-level prediction, please refer to Supplementary A. The cluster size of four patches was expected to best control the false-positive rate as shown in our early study[19], that is, the condition of continuously identifying four patches with cancer on WSI was used as the basis for determining the existence of cancer in WSI.

**Patient-level diagnosis**. Clinically, multiple WSIs may be obtained for one patient. The inference on the patient level was based on positive sensitivity, that is, if all WSIs from the same patient were identified as negative (no cancer), then the patient was negative, otherwise the patient was positive. At the patient-level diagnosis, Model-10%-SSL, Model-10%-SL, and the accurate SL model (Model-70%-SL) developed in our previous study[19] were compared.

**Methodology of lung and lymph node**. Two public data sets were used for the extended evaluation of SSL. The Dataset-Lung was from the LC25000[36], which consisted of 15,000 lung images (patches) including adenocarcinoma, squamous cell carcinoma, and benign tissue, and the number of each class was 5000 patches. The 20% images were used for testing, while the remaining 80% for training. A small number of labels (5%, 20% of 15,000 patches) together with a large number of unlabeled patches (75%, 60%, the labels were ignored) were used for SSL, while the 5%, 20%, and 80% labeled images were used for SL (Table 3).

The Dataset-Pcam was from PatchCamelyon dataset[37] including up to 300,000 patches of lymph node tissues, which had been split into training, validation, and testing sets. Meanwhile, the number of patches in training set was 262,144 patches, and the 1% and 5% of the patches was randomly extracted to simulate a small number of labeled data, while the remaining patches to simulate massive unlabeled data (the labels were ignored). The 36,728 patches in testing set were used for testing (Table 4), but 32,768 patches of validation set were not used.

Like CRC experiments, the base SL model was also Inception V3, and the mean teacher method was used for SSL. The 10% patches of the labeled training set were randomly selected for the validation set, which was used for the hyperparameter selection. This selection started from the hyperparameters of the CRC models, and tried the parameters nearby. The parameters were listed in Supplementary Tables 5 and 6.

The processing pipeline of the images from the lung and lymph node was like the CRC. Meanwhile, the patches were scaled to 299 × 299 based on the image interpolation. For SL of lung, the batch size was 64, the number of epochs was 500, the steps of each epoch were 100. The initial learning rate was 0.001, and the exponential decay was used with the decay rate was 0.99. The loss was the cross entropy with the L2 norm constraint, and the coefficient of L2 decay was equal to 0.0001. The early stopping was also used, where the patient was 50 epochs. For SSL of lung, the batch size was 32, the number of epochs, the steps, loss were the same with SL, but the learning rate was 0.0001, and remained the same. After the pre-training of 150 epochs, the early stopping was also used with the patience of 100 epochs. The smoothing coefficient of exponential moving average of the teacher network was set to 0.9.

Because the number of training patches in Dataset-Pcam was very large and the experiment time was very long, we continued to use the hyperparameters in lung experiments and tried to optimize them. For SL, the batch size, epochs, initial learning rate, decay rate, weights of L2 decay were the same with SL of lung, but the steps were changed to 300. We found the AUC and accuracy of Pcam-100%-SL can be compared to the benchmark provided by[37], so the hyperparameters were applicable. For SSL, the steps were 200. After the training of 80 epochs, the early stopping with patience 100 was used. The remaining hyperparameter were the same with SSL for lung models.

**Statistics and reproducibility**. To reduce the impact of random data set division and formally compare the performance of different methods, we applied the cross-validation and several statistical tests as following[56]. Taking CRC as an example, 70% of WSIs was randomly selected for the training set, and 30% of WSIs for testing set. Because the deep learning was time-consuming, the process of CRC dataset division was repeated eight times as well as lung cancer and lymphoma, and eight independent pairs of training set and testing set of every cancer were obtained. We repeated the training of the models such as Model-n%-SSL/SL, Lung-n%-SSL/SL or Pcam-n%-SSL/SL on the training set in the eight obtained data set pairs, and produced eight versions of each model, which were used for prediction on their testing set in the same data set pair respectively. The mean and standard deviation of the evaluation index (AUC or accuracy) were then calculated. The Wilcoxon-signed rank test was used to evaluate the significant difference between the two models based on their AUC or accuracy (sample size/group = 8). For patient-level evaluation, the Model-10%-SSL/SL, Model-70%-SL predicted the subjects from the twelve centers respectively, and the AUC of every model on each center was obtained. The Wilcoxon-signed rank test was also used to evaluate the significant difference of any two models based on their AUC on the centers (sample size/group = 12). Two-sided P values were reported for all statistical tests, and no adjustment was made.

**Reporting summary**. Further information on research design is available in the Nature Research Reporting Summary linked to this article.

## Data availability

The pathological images generated in this study have been deposited in the figshare database under accession code [https://doi.org/10.6084/m9.figshare.15072546.v1][57], where the images of Dataset-PATT, Dataset-Lung, and Dataset-Pcam can be used for patch-level retrain and retest. The independent patch-level testing set (Dataset-PAT) and 500 whole slide images in Dataset-PT have been provided with the source code for the patch-level and patient-level demo under accession code [https://zenodo.org/record/5524324#.YU09Ny-KFLY]. The remaining WSIs in Dataset-PT and Dataset-HAC can be obtained by contacting the corresponding author by Email [Kuan-Song Wang <375527162@qq.com>]. All data access in this study can only be requested by the researchers and for scientific research purposes. The data access requests will be processed in 10 business days. Source data are provided with this paper.

## Code availability

The source code generated in this study has been deposited in the zenodo database under accession code [https://zenodo.org/record/5524324#.YU09Ny-KFLY], including training and testing code of three cancers, and a demo[58]. The code is licensed under GNU (GNU's Not Unix) General Public License, and implemented by Python[59] and Tensorflow[60].

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

## Acknowledgements

This work was benefited by grant supports from the Emergency Management Science and Technology Project of Hunan Province (2020YJ004 and 2021-QYC-10050-26366 (G.Y.)), the National Natural Science Foundation of China (81972490 (K.-S.W.)), Natural Science Foundation of Hunan Province (2015JJ2150 (K.-S.W.)), the National Key Research and Development Plan of China (2017YFC1001103 and 2016YFC1201805 (H.-M.X.)), National Natural Science Foundation of China (81471453 (H.-M.X.)), Jiangwang Educational Endowment (H.-M.X.), the National Institutes of Health (R01AR059781, P20GM109036, R01MH107354, R01MH104680, R01GM109068, R01AR069055, U19AG055373, and R01DK115679 (H.-W.D.)) and the Edward G. Schlieder Endowment (H.-W.D.).

## Author contributions

Conceptualization, H.-W.D. and G.Y.; data curation, K.-S.W. and H.-M.X.; formal analysis, K.S., T.X. and R.-Q.M.; investigation, K.S., T.X. and X.-H.M.; methodology, G.Y.

and H.-W.D.; project administration, K.-S.W., H.-M.X. and H.-W.D.; resources, K.-S.W.; software: G.Y., K.S. and T.X.; supervision, H.-W.D. and H.-M.X.; writing original draft preparation, Y.G., K.S., T.X. and R.-Q.M.; writing—review and editing, Y.G., C.X., X.-H.S., C.W., T.X., R.-Q.M. and H.-W.D. All authors have read and agreed to the published version of the manuscript.

## Competing interests

The authors declare no competing interests.
