## [Peer Review File · Nature Communications]

Reviewers' Comments:

Reviewer #1:

Remarks to the Author:

In this manuscript, the authors describe a method for detection of colorectal cancer tissue in digitized histology images. They claim that the novelty of their method is a reduced need for annotated training data compared to other approaches. Furthermore, they claim that this will ultimately improve

The main positive point in this study is the high sample size. Although the number of cases in this study is certainly impressive, there are a number of major shortcomings in this article which compare very unfavourably to other studies published in Nature Comms.

- tumor detection of CRC tissue is not a clinically relevant benchmark task. Compared with detection of prostate cancer in digitized histology images (Campanella et al., Nat Med), CRC is very easily detectable even for non-experts. A pathologist only needs seconds to spot a typical CRC case, even without a microscope. There is no indication in this article that only difficult cases were analyzed, so the task at hand seems trivial from a clinical point of view.

- tumor detection in CRC tissue has been long solved with near-perfect performance. Even with classical machine learning (no AI), CRC tumor tissue can be almost perfectly detected in histology images (Kather et al., Sci Rep 2016). So, the task at hand is already solved and the authors do not show that their new method outperforms these previous methods (which like the example above, needed <1000 annotated tiles).

- the claims of the article are too strong (eg last section of the abstract). The authors oversell their findings of this tumor detection study.

Additional comments:

- In many places throughout the manuscript, the text is carelessly written. For example, in the abstract, AUC is sometimes expressed between 0 and 1, sometimes between 0% and 100%. Also, abbreviations are not always defined such as "AUC" in the abstract.

- generally, accurate detection of tumor tissue on patient level is much easier than on patch level. A very imperfect patch-based classifier can yield reasonable patient-level predictions. this is not adequately discussed in the article.

- clinically interesting tasks such as detection of intraepithelial neoplasia are not adequately discussed

- The "results" section is very short and gives insufficient detail

- the figures are of poor quality (inconsistent color map, poor choice of font, ...)

- I did not find any reference to the source codes in the main article. This is not acceptable as most papers in this field make all source codes available.

- The paper does not adhere to the STARD-AI guidelines (<https://www.nature.com/articles/s41591-020-0941-1>)

Reviewer #2:

Remarks to the Author:

** Main claims:

Most deep-learning approaches rely on training of large and properly labelled datasets. Obtaining those labels can be difficult, especially in histopathology where slides are large and such annotations can be time-consuming to generate for pathologist.

In their manuscript, Yu et al. study the performance of a semi-supervised deep-learning approach that could reduce the need for large amount of labelled data. They show that if only 1/7 of their training set is labeled, the performance is almost as good as when the training set is fully labeled.

** Interest:

I believe such study can really be interesting to the field as obtaining large amount of annotated data is known to be an issue... providing the code will ever be made available and be well documented to ensure reproducibility, which doesn't seem to be the case yet.

** Are the claims convincing? If not, what further evidence is needed?

The number of images used here is impressively high (13,111 whole slide images from 8,803 subjects) and are expected to represent a wide diversity of patients and image acquisition/specimen preparation since they come from 13 different centers. However, there is a lack of information regarding the method itself and the architectures used (see comments below)

** Are the claims appropriately discussed in the context of previous literature?

There are two aspect that need to be referenced:

- Performance of the method for this particular application

Most references are present in Supp table 4, but that table does not seem to be referenced or discussed anywhere.

- Discussing the context of un-supervised or semi-supervised approaches in WSI analysis

This second point is poorly discussed (point 4 of main comments below)

** Have they provided sufficient methodological detail that the experiments could be reproduced?

Should the authors be asked to provide further data or methodological information to help others replicate their work? (Such data might include source code for modelling studies, detailed protocols or mathematical derivations)

Not enough – I don't see any code or README file that would detail the steps to do to reproduce the results

** Comments:

1- Percentages in table 3 is difficult to understand (how the percentages were computer or what they mean). I would expect to find 100% of tiles if I add them up by lines or columns but it doesn't match. Why are the first two columns in % and the last one number of patches? The name of the "unused label" is misleading. Did you mean unused or unknown? Because in the SSL model, I thought they were used. Putting also the number of tiles might be helpful

2- Figure 2: Please, convert y label to English

3-"Human-AI competition"

3a- Please describe under which circumstances has the pathologists worked in method (were there time constraints, what other data did they have compared to those they used in "real" clinical case? [only the images or the patient history, etc... as well...])

3b- For the AI, are these per patch, per slide or per patient AUCs?

3c- (optional suggestion) You have established before the 10%-SL model performs less well. I would remove it from this graph (fig 4) so you can better adjust the limits of the y-axis.

3d- the legend says "using Dataset-HAC." But you show for 9 other datasets as well, no?

3e- What is the difference between the 6 pathologists here and those used to select the slides and their true label? What is the probability that some slides were mis-labelled? When the 6 pathologists made mistakes, what proportions of those mistakes were on the same slides and how did the AI perform for those particular slides misclassified by the pathologists? Do the AI and the pathologist tend to make mistake on the same slides?

3f – It looks like the pathologists had to decide only whether tumor cells were present or not, right? What was the source of mistakes? Small tumors that were missed, or other cases?

4- I appreciate the comparison in Supp Material F (though the section should be referenced in the

main text). However, the literature on the more general subject of semi or un-supervised approach for annotation of WSI in general is poorly represented in the paper. Please consider discussing this approach versus other existing approaches that were published to achieve similar goals (for example, Quiros A.C., Murray-Smith R., Yuan K. Learning a low dimensional manifold of real cancer tissue with PathologyGAN. 2020 and others)

5- Very important: there is no code available to assess the reproducibility of the work. The code should be made available to the reviewer with a well detailed README file. If, for confidentiality reasons, the code cannot be shared with the reviewers, then, at least the README file should be shared to ensure that all the details required are there to check future use of the code is possible.

6- I would be interested in seeing how long it takes to train the network for the different types of models. When do they converge and how was the convergence epoch determined?

** Comments on Supp Method A:

7- At the end of "Digitization and annotation" section of Supp Method A, specify more clearly what happens to the other patches. For example, "In model-5%-SSL [...] small proportions (i.e., 5% and 10%) of total patches (62,919) and masked label information for the remaining patches". The remaining patches mean 95 and 90% respectively, right? What do you mean by "masked label information"? In table 3, it looks like 30% was used for test and 65% un-used. Is that correct? I don't see how it fits with what's said above.

8- For the other model, "In model-5%-SL, model-10%-SL and model-70%-SL, we used supervised training with 5%, 10%, 70% of the total 62,919 patches", what happens to the remaining 95, 90 and 30%? Are they used as PATT test as in Table 3?

9- "Because the WSI is very large (>50,000 pixels), the patch-level models were trained to recognize cancerous probability, and all the patch-level results on a WSI were combined to infer the cancerous probability of the WSI. The flow chart is shown in Figure 1."
How were they combined exactly?

10- "We tested the known CNNs, such as VGG16, ResNET V1 and V2, Inception V1-V4, Mobilenet, etc., and found that InceptionV3 [16] achieved most consistent results on many datasets. Therefore, we used Inception V3 as the baseline model."

- Please, do not use "etc". Be specific which ones were used

- You state "InceptionV3 [16] achieved most consistent results on many datasets." Since you've done those tests, please show the results in a (supp?) figure to support that observation

11- "The patch-level models included supervised and semi-supervised versions. The input patch size was scaled to 299x299"

Before, you said you extracted 300x300 pixel patches. Was it scaled from 300x300 to 299x299? This sounds weird. Why not extract the patches directly at 299x299?

12- "The semi-supervised version was based on the mean teacher method [14], where two Inception V3 were trained, one as student and the other as teacher." Please reference Supp material C. Also, the paper itself would benefit from a clearer description of this architecture.

13- "The Inception V3 adopted the pre-trained model on ImageNet database, and was deeply fine-tuned."

What do you mean? Did you do transfer learning, or do you mean something else? Also, what fine-tuning was done exactly?

14- "The training cycle was 100 epochs, each epoch included 100 steps."

And what was the batch size in each step? Since you say "because of the imbalance between the labeled and unlabeled data, we maintained the same proportion of labeled and unlabeled patches in each mini-batch.", I guess it means some labelled patches were used several times and unlabeled patches once. Is that the case? In this particular case then, how is "1 epoch" defined exactly then?

15- "Clustered-based WSI inference" section: what if some patches are at the edge of two regions, one made of non-cancerous cells, the other of cancerous cells?

16- Supp table 4: for the "other cancers" section, please specify for each of the which cancer

** Other minor comments:

17- Fig 1 would benefit from more details on the legend (for example, below the models on the right is written things like "5%/10%-SL" - Please explain in the legend that it is the model name). Since the Fig 1 is referred before explaining the differences or the associated tables, legend should be more complete.

18- PAT – table 1 and 2: why are the number of subjects/tiles unknown? How were those obtained?

19- Supp Method A: "From the 842 WSIs in Dataset-PATT, two pathologists manually selected some representative patches,"

How was it done? Did they draw selection directly on the WSI from which patches were extracted afterwards (in which case, which software package was used), or were the patches extracted first, and annotated after by the pathologists (in which case, how were the 100 patches max per WSI pre-selected, and why are they ~60,000 patches in the end instead of 842x100 patches?)?

20- Only Supplementary A is referred in the main text, others are not.

21- Only 2 scenarios were tested (5%-SSL and 10%-SSL). I understand it is impossible to test all combination, but can the authors comment (or do a kind of power analysis) on, in general, what would be the optimal percentage of labeled data needed or how to find it in an efficient way? Also, since you are not limited to the number of unlabeled data, would the performance be even more improved if you were to use a much higher number of unlabeled patches?

** As a conclusion:

the paper overall is quite interesting. I think developing architectures or strategies that allow good prediction with few labels is important.

Strengths of the paper:

- The number of slides used is high and the dataset comes from a very large number of different institutions, supporting therefore well the conclusions of the paper
- Comparison with 6 pathologists is performed
- The application is important to the field to reduce the number of labeled data needed

Major weaknesses that would need to be addressed to make the approach more sound/robust:

- many sections would benefit from major clarifications. Also making sure, in the workflow, that dataset, acronyms and data are well explained the very first time they are mentioned in the text would help the flow.
- Description of the architecture
- Code availability (or README file) missing

Reviewer #3:

Remarks to the Author:

The proposed work investigates semi-supervised deep learning for the classification of colorectal cancer on pathological images. A large dataset with 13,111 whole slide images was used for the investigation, and the results show that, with a relatively small number of annotated training

samples, semi-supervised classification can be comparable to fully-supervised learning with a large number of annotated training samples. These results suggest that semi-supervised learning can reduce the cost of annotations of colorectal cancer and/or allow decent classification performance when the annotations are scarce.

The problem of scarce annotations in medical imaging is in general an important one to address and indeed requires attention. The proposed work could make an impact. However, semi-supervised deep learning has already been explored in many studies for medical imaging (for example, see [1] for a review of relevant studies). Although the application to colorectal cancer is new, the proposed work uses the well-established mean teacher method, which is quite commonly used for medical image classification (see [2] and [3] for example). Thus, the novelty in terms of application is incremental, and the technical contribution is little.

As for the experiments, it is a strength that a relatively large number of images from different centers were included. However, considering the limited novelty in terms of application and methodology, I would suggest experiments with a wider range of diseases to increase the impact of the proposed work. Can the authors test three or four different diseases on whole slide images, and then conclude that semi-supervised deep learning is beneficial for general image classification based on whole slide images?

The authors claim that "how to translate the patch-level prediction to WSI and patient level diagnosis is not trivial". Please elaborate why.

I am also somewhat confused about the generation of patch-level annotations. While I believe the patient-level annotation is determined by whether cancer exists in the image of the patient, for the patches, do we need experts to look at each patch and determine if cancer exists in the patch too? If that is the case, the annotation can be quite laborious. Please correct me if I understand it incorrectly and clarify.

In Table 4, row 6 is a repeat of row 4. Can there be a more efficient way of presentation without such repetitions?

There are some grammatical errors. For example, in line 192, "models currently relies" -> "models currently rely". Please double check thoroughly.

References

- [1] Cheplygina et al., Not-so-supervised: A survey of semi-supervised, multi-instance, and transfer learning in medical image analysis, *Medical Image Analysis*, 2019.
- [2] Su et al., Local and Global Consistency Regularized Mean Teacher for Semi-supervised Nuclei Classification, *International Conference on Medical Image Computing and Computer-Assisted Intervention*, 2019.
- [3] Liu et al., Semi-supervised Medical Image Classification with Relation-driven Self-ensembling Model, *IEEE Transactions on Medical Imaging*, 2020.

Response to reviewers' comments

Reviewer #1:

In this manuscript, the authors describe a method for detection of colorectal cancer tissue in digitized histology images. They claim that the novelty of their method is a reduced need for annotated training data compared to other approaches. Furthermore, they claim that this will ultimately improve.

The main positive point in this study is the high sample size. Although the number of cases in this study is certainly impressive, there are a number of major shortcomings in this article which compare very unfavourably to other studies published in Nature Comms.

R1-Q1:

Tumor detection of CRC tissue is not a clinically relevant benchmark task. Compared with detection of prostate cancer in digitized histology images (Campanella et al., Nat Med), CRC is very easily detectable even for non-experts. A pathologist only needs seconds to spot a typical CRC case, even without a microscope. There is no indication in this article that only difficult cases were analyzed, so the task at hand seems trivial from a clinical point of view.

We sincerely thank the reviewer for the positive comments on the significance of our research.

When using a large number of labeled datasets, cancer detection via AI may achieve good results for different cancers, ranging from CRC (colorectal cancer) to other types of cancer. However, clinically labeled data are always scarce, and in many cases, it may become impractical to collect sufficient data that are ready for AI analyses. Therefore, it is of great value to develop new tools and pipelines that can efficiently use limited amount of labeled data, augmented by the vast majority of unlabeled data in most practical cases, to support data-driven clinical decisions including cancer detection.

Hence, our major research goal here is not to prove that artificial intelligence can be used for diagnosis for CRC, but rather to use CRC as an example of a specific cancer type to demonstrate how to build a medical artificial intelligence system with only a small amount of labeled data plus most of other unlabeled data. The reason we chose to study CRC here in this study is because we are fortunate to have access to a large amount of CRC data in which pathological diagnosis of every whole slide image has been clearly and consistently labeled. We have developed a recognition system for CRC using a supervised learning (SL), which achieved one of the highest diagnostic accuracies in the AI field when applied to cancer diagnosis (Wang Kuan-Song, BMC Medicine, 2021, 19(76):1-12). However, our earlier method was built upon learning from 62,919 labeled patches from 842 subjects, which were carefully selected and extensively and laboriously labeled and double checked for consistency by pathologists.

Hence, we can perform the investigation, in this study, by using various proportions of the data as labeled and other data as unlabeled (with their labels hidden) for investigation of the proposed semi-supervised learning and its comparison with supervised learning when the labels for all the patches are known. Compared with existing methods most of which require exceedingly large amount of labeled data, we provided and demonstrated a semi-supervised learning framework to speed up the deployment of medical artificial intelligence by drastically reducing the amount of labeled data needed. For clarification, we have revised the introduction of the manuscript to clearly state the motivation, significance and potential application of this proposed work and approach:

Paragraph 3-4 in Introduction section.

While supervised learning with massive labeled data can achieve high diagnostic accuracy, the reality is that we often have only a small amount of labeled data and a much larger amount of unlabeled data. The semi-supervised learning (SSL), a method that leverages both labeled and unlabeled data are supposed to provide a low-cost alternative to the supervised learning (SL) in terms of the requirement of the laborious sample labeling.

Paragraph 6 in Introduction.

To further confirm that SSL can achieve excellent performance on pathological images and further demonstrate our main point that a reliable medical AI can be built with a small amount of labeled data, we evaluated the performance of SSL on two other types of cancer (lung cancer and lymphoma).

R1-Q2:

The claims of the article are too strong (eg last section of the abstract). The authors oversell their findings of this tumor detection study.

We thank the reviewer for the comment! We have toned down our claims and modified the abstract and article to reflect these changes in the last section of the abstract:

We reported that SSL can achieve excellent performance through a multi-center study. Because SSL dramatically reduces the cost of pathological image annotation, it has great potential to effectively build pathological artificial intelligence (AI) platforms.

R1-Q3:

In many places throughout the manuscript, the text is carelessly written. For example, in the abstract, AUC is sometimes expressed between 0 and 1, sometimes between 0% and 100%. Also, abbreviations are not always defined such as "AUC" in the abstract.

We thank the reviewer for pointing this out! We regret that we have made these mistakes in our original manuscript. We have carefully revised the manuscript. We now describe AUC in a consistent way (between 0 and 1). We also defined all the abbreviations before using them in the revised manuscript.

R1-Q4:

Generally, accurate detection of tumor tissue on patient level is much easier than on patch

level. A very imperfect patch-based classifier can yield reasonable patient-level predictions. this is not adequately discussed in the article.

We thank the reviewer for this helpful and insightful comment! We have revised our manuscript and included more discussions about our observations on the patch- and whole slide image (WSI)/patient-level predictions. Because WSI is too large as input for the model, the patient level model is always developed at the patch level first. So far, we and other groups have not been able to develop patch-level models of 100% accuracy. Usually, if the sensitivity of the patch-level model is not too low, the sensitivity of WSI/patient may be increased because if only one positive patch of a large number of positive patches on WSI is found, the WSI may be consider as cancer positive. However, if its sensitivity is too low, none of the true positive patches could be found. If we make the sensitivity of patch-level model increased to improve the sensitivity of WSI/patient, its specificity will decrease accordingly. The patch-level model with low specificity will consider some negative patches as positive, so false positive errors of WSIs will be accumulated on multiple patches of WSIs. Therefore, how to derive a satisfactory patient-level inference based on an imperfect patch-level model is not an easy task and requires further studies. We also discussed the issue great detail in our previous study (Wang Kuan-Song, BMC Medicine, 2021, 19(76):1-12) and supplementary A and B.

We discussed the issue in paragraph 5 in the introduction:

Because we and other groups have not been able to develop perfect patch-level models, the errors at patch-level may be easily magnified on WSI level diagnosis. For example, even though the imperfect patch-level model may yield reasonable prediction on positive (cancerous) WSIs, it also may yield high false positive errors on the negative (non-cancer) WSIs, because the false positive errors at patch level will accumulate due to the testing of multiple patches in WSI. However, the patient-level diagnosis is required in clinical applications of any AI system for cancer diagnosis.

R1-Q5:

Clinically interesting tasks such as detection of intraepithelial neoplasia are not adequately discussed.

Please see **R1-Q1** for the clarification of the motivation and significance of the current work.

Although we agree with the reviewer that intraepithelial neoplasia can be a clinically interesting case, we have decided to focus on CRC mainly due to the availability of the labelled data as we stated earlier.

In the revised manuscript, we did add two other cancer types, namely lung cancer and lymphoma, where we applied our methodology of SSL to further demonstrate the application and power of our SSL methods to cancer prediction in different cancer types.

R1-Q6:

The "results" section is very short and gives insufficient detail

We thank the reviewer for pointing this out! We regret not including sufficient details in this important section. We have thus revised the manuscript as follows.

We added more details in Result section.

(1) SSL vs SL CRC recognition at patch level

The area under the curve (AUC) distribution on Dataset-PATT and Dataset-PAT were shown in Figure 2. Model-5%-SSL was superior to Model-5%-SL (AUC of both Dataset-PATT and Dataset-PAT, 0.75 confidence interval: 0.90 ± 0.06 vs. 0.84 ± 0.07 , P value=0.02, Wilcoxon signed rank test). Model-10%-SSL was also significantly better than Model-10%-SL (AUC: 0.98 ± 0.01 vs. 0.92 ± 0.04 , P value=0.0004). These results indicated that when approximately 3,150 or 6,300 patches were labeled, the SSL method was always better than SL.

The performance of Model-10%-SSL had no significant difference with that of Model-70%-SL (AUC: 0.98 ± 0.01 vs. 0.987 ± 0.01 , P value=0.134). This observation indicated that there was no significant difference between the SSL method (6,300 labeled, 37,800 unlabeled) and the SL (44,100 labeled). Visual inspection (Supplementary Figure 2) confirmed that that Model-10%-SL could not really find the locations of cancer in the patches, while the locations of cancer by Model-10%-SSL and Model-70%-SL were highly matched.

(2) Patient-level CRC recognition

Model-10%-SSL had a significant improvement over Model-10%-SL (Average AUC: 0.974 vs. 0.819, P value = 0.0022) on patient-level prediction in the multi-centers scenario. The average AUC of Model-10%-SSL was slightly lower than, but comparable to, that of Model-70%-SL (average AUC: 0.974 vs. 0.980, P value = 0.117). Among the 7 datasets (XH-dataset-PT, XH-dataset-HAC, PCH, TXH, FUS, SWH, TCGA, 11,290 WSIs), the AUC difference of Model-10%-SSL and Model-70%-SL was smaller than 1.6%. In particular, on the largest dataset, XH-dataset-PT (10,003 WSIs), the AUCs of Model-10%-SSL and Model-70%-SL were close with 0.984 vs. 0.992. On the datasets of HPH, SYU, CGH and AMU (501 WSIs), the AUCs of Model-10%-SSL were even higher than that of Model-70%-SL.

In the data from GPH, and ACL (392 WSIs), the performance of Model-10%-SSL was lower than that of Model-70%-SL (AUC DIFF>0.222). It is worth noting that Model-10%-SSL generally achieved good sensitivity, which proved practically useful for the diagnosis of CRC. Visual inspection in Supplementary Figure 3 showed the cancer patches identified by Model-10%-SSL and Model-70%-SL were the true cancer locations on WSIs.

(3) Human-AI CRC competition

We recruited six pathologists with 1-18 years of independent experience (Supplementary Table 4). They independently reviewed 1,634 WSIs from 10 data centers (Dataset-HAC, Figure 4) with no time limit and diagnosed the cancer solely based on WSIs (i.e., no other clinical data were used). We ranked pathologists, Model-10%-SSL and Model-70%-SL. The average AUC of model-10%-SSL was 0.972, ranked at the 5th, which was close to the average AUC of pathologists (0.969). The

sensitivity of Model-10%-SSL was 0.977, showing an excellent detection ability of cancer (Supplementary Table 5).

(4) Comparison with related research

We compared our methods with seven existing CRC detection methods [35,40-45], and five other cancers (lung, ductal carcinoma, breast, prostate, basal cell carcinoma) detection methods [46-50] (Supplementary Table 10). The 6 of 7 CRC detection methods had an AUC ranging from 0.904 to 0.99 based on SL. Besides, Shaw et al [35] used cancer and normal patches in 86 CRC WSIs to develop a SSL detection method, and used the test set of 7,180 patches in 50 WSIs with colorectal adenocarcinoma, all from one data center, with the best accuracy of 0.938 confirming the potential of SSL on patch-level. In this study, we showed the advantages of the SSL method with 162,919 patches and 13,111 WSIs at both patch and patient levels from multiple independent centers, attesting to the robustness and general utility of the SSL model we developed, where the Model-10%-SSL is comparable to the recent SL models [19]. Besides, Lung-20%-SSL is also comparable to the SL of Coudray et al for lung cancer detection [46].

R1-Q7:

The figures are of poor quality (inconsistent color map, poor choice of font, ...)

We thank the reviewer for pointing this out. We have improved the quality of the figures in the revised manuscript as suggested, including Figure 1, Figure 2, Figure 3, and Figure 4.

R1-Q8:

I did not find any reference to the source codes in the main article. This is not acceptable as most papers in this field make all source codes available.

We thank the reviewer for pointing this out. We have uploaded the source code in GitHub. The code and data can be found from www.github.com/csu-bme/pathology_SSL, including a detailed README.

R1-Q9:

The paper does not adhere to the STARD-AI guidelines (<https://www.nature.com/articles/s41591-020-0941-1>)

We thank the reviewer for pointing to this resource.

Thank you for your suggestion on manuscript improvements.

STARD-AI in the provided the link (<https://www.nature.com/articles/s41591-020-0941-1>) discussed how to improve the usage of labeled data and unlabeled data, not overly exaggerating the role of AI in medicine.

We also searched and identified STARD (Standards for Reporting of Diagnostic Accuracy Studies) published in 2015 which addresses issues include unclear methodological interpretation (e.g., the use of external validation datasets, complexities of datasets and comparison to human performance) and the lack of standardized nomenclature (e.g., the definition of a ‘validation dataset’), as well as the heterogeneity of outcome measures (e.g., area under the receiver operating characteristics (AUROC), sensitivity, positive predictive value and F1 score). Because STARD was not designed to address the issues and challenges raised by AI-driven modalities, the STARD-AI Steering Group is preparing an AI-specific extension to the STARD statement (STARD-AI) that aims to focus upon the specific reporting of AI diagnostic accuracy studies. According to the link provided, they anticipated that the publication of the final recommendations in late 2020.

We prepared this article in the first half of 2020, so we did not see the above webpage. We check the reference 5 in the provided link, and try look for STARD-AI in <https://www.equator-network.org/library/reporting-guidelines-under-development/reporting-guidelines-under-development-for-other-study-designs/#STARDAI>. However, the newest message is: They plan to publish the reporting guideline, as an open-access document, in several journals, in the first quarter of 2021. We then searched STARD-AI by Google, but no more information found.

Although the STARD-AI is not available now, our current study generally meets of the STARD criteria. For example: when designing the experiment, we used a huge CRC data set, which was collected by ourselves and other centers, including some online databases. Two other cancer datasets were also included for external validation. The external validation dataset is extensive and heterogeneous (thus offering an opportunity for generality assessment). We also used some common index for diagnosis accuracy, such as sensitivity, specificity, accuracy, and AUC, which have most comprehensively shown the performance of the method. The source code has been uploaded to GitHub: www.github.com/csu-bme/pathology_SSL for the test and assessment by other researchers.

Reviewer #2:

Most deep-learning approaches rely on training of large and properly labelled datasets. Obtaining those labels can be difficult, especially in histopathology where slides are large and such annotations can be time-consuming to generate for pathologist.

In their manuscript, Yu et al. study the performance of a semi-supervised deep-learning approach that could reduce the need for large amount of labelled data. They show that if only 1/7 of their training set is labeled, the performance is almost as good as when the training set is fully labeled.

R2-Q1:

I believe such study can really be interesting to the field as obtaining large amount of annotated data is known to be an issue... providing the code will ever be made available and be well documented to ensure reproducibility, which doesn't seem to be the case yet.

Are the claims convincing? If not, what further evidence is needed)

Many thanks for the comment! We have uploaded the source codes and data in GitHub at www.github.com/csu-bme/pathology_SSL. Other researchers can repeat our experiments through the provided source code for reproducibility.

R2-Q2:

The number of images used here is impressively high (13,111 whole slide images from 8,803 subjects) and are expected to represent a wide diversity of patients and image acquisition/specimen preparation since they come from 13 different centers. However, there is a lack of information regarding the method itself and the architectures used (see comments below)

Thanks for the comment! We have provided more detailed description of the methodological framework as detailed below. More information of method and architectures has been added to Supplementary A.

(1) we firstly described the datasets of five training models (patch-level SSL and SL section in Supplementary A):

The Dataset-PATT was randomly divided into training set and testing set according to the proportions showed in Table 3, and the patches from the same subject would not be in different sets, to ensure independence of the different data sets.

Five patch-level models (two SSL, three SL) were trained using labels of different portions of these patches (Table 3). In the training of model-5%-SSL and model-10%-SSL, we used SSL and kept labels for small proportions (i.e., 5% and 10%) of the total patches (62,919) and masked label information for the remaining patches (65% and 60%). In the training of model-5%-SL, model-10%-SL and model-70%-SL, we used SL with 5%, 10%, 70% of the total 62,919 patches. Because training a deep learning model is time-consuming, for illustration, we repeated the process 8 times and calculated 0.75 confidence interval. The Model-70%-SL was trained on the same number of

labeled patches with SL as in our previous study [2]. The Dataset-PAT was used as an independent test.

(2) We then described the algorithm pipeline, patch-level SSL and SL in Supplementary A.

Because WSI is very large (>50,000 pixels), the patches in a WSI were firstly extracted, and the patch-level models were trained to derive cancerous probability at patch-level. Finally, all the patch-level results on a WSI were combined to infer the cancerous probability of the WSI/patient. The flow chart is shown in Figure 1.

Patch-level SSL and SL

The patch-level models included SL and SSL versions. For SL, the patches from the WSIs were input to the CNN. Our previous work tested some known CNNs, such as VGG16, ResNET V1 and V2, Inception V1-V4, and found that Inception V3 [1] achieved most consistent results on the CRC datasets [2]. Therefore, we used Inception V3 as the baseline model of SL. The patch size we labeled was 300×300, so we used the bilinear interpolation method to scale the patch size to 299×299, which is the default input size of Inception V3. The top output layer was removed, and the output category was modified to two (cancer or non-cancer).

The SSL version was implemented based on the mean teacher method [3], where two Inception V3 were trained, one as student and the other as teacher, which is one of SSL method (Supplementary Figure 1). The student network uses SL and requires inputted patches, which include a small number of patches with labels and large number of unlabeled patches. For the labeled patches, the cross-entropy of the predicted and real label was calculated as the classification cost. For unlabeled patches, teacher network provided the pseudo labels, and the mean square of the predicted labels and pseudo labels was calculated as consistency cost. The weighted sum of consistency cost and classification cost, as the total cost, was used for the student network training. In this study, the two networks were performed on the same architecture with SL, i.e., Inception V3.

(3) More details of training were added in network training at patch level section in Supplementary A:

We initialized the Inception V3 with a pre-trained model on ImageNet database, and then trained it on pathological images. During training, all layers of Inception V3 were updated, which was deeply fine-tuned. We used the same preprocessing in protocols we used earlier [2]. All background patches without any cell tissue were removed. After data augmentation (image zoom, flip, color change), the grayscale of each pixel was normalized to [-1,1].

For each model, we adopted a general strategy where the one-tenth of the labeled training set was taken out as the validation set for hyperparameters selection. The optimal hyper-parameters with highest accuracy in the validation set were selected for training the models. The parameters were listed in supplementary Table 2.

In the SSL, because of the imbalance between the labeled and unlabeled data, we maintained the same proportion of labeled and unlabeled patches in each mini-batch of 128 patches. The optimizer was Adam. The training period was 500 epochs, and each epoch included 100 steps. If the

accuracy on validation set can't be improved for 80 consecutive epochs, the early stopping [22] was applied. In order to prevent the training from ending prematurely, 50 epochs for pre-training were executed before the early stopping. L2 decay was used and the decay coefficient was set to 0.0001. The teacher network was initialized with the student network. The student network would update the weights in each step, but the teacher network used exponential moving average to update the weights after one epoch end. The smoothing coefficient was set to 0.95.

In SL, the learning rate was 0.001, and the exponentially decay was used with the decay rate 0.99. The number of epochs was 500, the steps per epoch was 100. The early stopping with patience 50 was also applied. The coefficient of L2 decay was 0.0001, and the batch size was 64.

(4) More details were added for clustered-based WSI inference (clustered based WSI inference section in Supplementary A):

Because the accuracy of patch-level models cannot be 100%, there were serious false positives in WSI predictions if any patch in the WSI was identified as positive (cancer) and used as a criterion for predicting the WSI cancerous status. Intuitively, because the tissues in WSI were continuous, the area with cancer should be distributed continuously and included several continuous positive patches. This intuition had been used to effectively control the false-positive of functional magnetic resonance images [4]. We designed a simple clustering-based inference method. If some continuous patches were identified as having cancer by patch-level model, the cancer may indeed exist on WSI. The cluster size of four patches was expected to best control the false-positive rate as shown in our early study [2], that is, the condition of continuously identifying 4 patches with cancer on WSI was used as the basis for determining the existence of cancer in WSI. For statistical analysis on patient-level prediction, please refer to Supplementary B.

R2-Q3:

Are the claims appropriately discussed in the context of previous literature?

There are two aspect that need to be referenced:

- Performance of the method for this particular application

Most references are present in Supp table 4, but that table does not seem to be referenced or discussed anywhere.

Thanks! We added the discussion on the methods of this particular application. (Pages 3 in Introduction):

While SL with massive labeled data can achieve high diagnostic accuracy, the reality is that we often have only a small amount of labeled data and a much larger amount of unlabeled data in medical domains. Although unsupervised learning does not require any labeled data, its performance is still limited currently [20,21]. There are some other approaches for learning on the small amount of labeled data. For example, in transfer learning, the network is firstly trained in a big dataset of source domain, and then trained in labeled medical images. However, the number of labeled images needed is still quite large [22, 23]. The generative adversarial networks (GAN) can generate large amount of data by learning the style from a limited dataset [24, 25]. These approaches may improve accuracy, but they only used limited labeled datasets, and large amounts of unlabeled data do appear

in medical domains and clinical settings. Moreover, it would be difficult for GAN to simulate all possible features of the disease based on limited samples.

Supplementary Table 4 in the last submission (Now Supplementary Table 10 in this revised manuscript, because some new tables were added during revisions) has been discussed in the Results section:

We compared our methods with seven existing CRC detection methods [35,40-45], and five other cancers (lung, ductal carcinoma, breast, prostate, basal cell carcinoma) detection methods [46-50] (Supplementary Table 10). The 6 of 7 CRC detection methods had an AUC ranging from 0.904 to 0.99 based on SL. Besides, Shaw et al [35] used cancer and normal patches in 86 CRC WSIs to develop a SSL detection method, and used the test set of 7,180 patches in 50 WSIs with colorectal adenocarcinoma, all from one data center, with the best accuracy of 0.938 confirming the potential of SSL on patch-level. In this study, we showed the advantages of the SSL method with 162,919 patches and 13,111 WSIs at both patch and patient levels from multiple independent centers, attesting to the robustness and general utility of the SSL model we developed, where the Model-10%-SSL is comparable to the recent SL models [19]. Besides, Lung-20%-SSL is also comparable to the SL of Coudray et al for lung cancer detection [46].

R2-Q4:

Discussing the context of un-supervised or semi-supervised approaches in WSI analysis This second point is poorly discussed (point 4 of main comments below)

We add some discussion on un- and semi-supervised approaches in Paragraphs 3-4 of the Introduction section:

While SL with massive labeled data can achieve high diagnostic accuracy, the reality is that we often have only a small amount of labeled data and a much larger amount of unlabeled data in medical domains. Although unsupervised learning does not require any labeled data, its performance is still limited currently [20,21].

The SSL (semi-supervised learning), a method that leverages both labeled and unlabeled data is supposed to provide a low-cost alternative in terms of the requirement of the laborious and sometimes impractical sample labeling [26, 27]. Although SSL can improve accuracy of natural images, its performance on medical images is unclear. Recently, some studies were proposed to determine whether SSL based on a small amount of labeled data and a large amount of unlabeled data can improve medical image analysis [28-30], such as object detection [31], data augmentation [32], image segmentation [33,34]. However, only very limited few studies have investigated if SSL can be applied to achieve satisfactory accuracy in pathological images [35], where on a small data set of 115 WSIs, an SSL method of CRC recognition can achieve best accuracy of 0.938 only at 7,180 patches of 50 WSIs from one data center, suggesting the potential of SSL for diagnosis on patch-level.

R2-Q5:

Have they provided sufficient methodological detail that the experiments could be reproduced? Should the authors be asked to provide further data or methodological information to help others replicate their work? (Such data might include source code for modelling studies, detailed protocols or mathematical derivations)

Thank you for your recommendation. We extend more methodological information on Supplementary A and B. In Supplementary A. Firstly, we described the details of the dataset and data annotation of patches and WSIs. Secondly, we added the training details, and provided all the hyper-parameters. Thirdly, we described the presented networks, model setting and algorithm pipeline. In Supplementary B, we provided the statistical analysis on patient-level prediction.

R2-Q5a:

Not enough – I don't see any code or README file that would detail the steps to do to reproduce the results

We have uploaded the source codes in GitHub (www.github.com/csu_bme/pathology-SSL). The readme file was also included in the link. The results in the article are reproducible.

R2-Q6:

Percentages in table 3 is difficult to understand (how the percentages were computer or what they mean). I would expect to find 100% of tiles if I add them up by lines or columns but it doesn't match. Why are the first two columns in % and the last one number of patches? The name of the "unused label" is misleading. Did you mean unused or unknown? Because in the SSL model, I thought they were used. Putting also the number of tiles might be helpful

Thanks for the suggestion! We revised the table to improve ease of understanding. The "unused label" means the patches are used in SSL model, but their labels are not used so they are used as if they are unlabeled. The revised Table 3 is now as follows.

Table 3. Training and testing set for CRC patch-level models

Model	Class	Dataset-PATT			Dataset-PAT
		Training set		Testing set	
		Labeled	unlabeled ^f		
Model-5%-SSL	Cancer	1,645	21,390	9,828	14,317
	Non-cancer	1,505	19,560	8,991	85,683
	Total	3,150/5% ^a	40,950/65% ^b	18819/30% ^c	100,000
Model-10%-SSL	Cancer	3,290	19,745	9,828	14,317
	Non-cancer	3,010	18,055	8,991	85,683
	Total	6,300/10%	37,800/60% ^d	18819/30%	100,000
Model-5%-SL	Cancer	1,645	-	9,828	14,317
	Non-cancer	1,505	-	8,991	85,683
	Total	3,150/5%	-	18819/30%	100,000

Model-10%-SL	Cancer	3,290	-	9,828	14,317
	Non-cancer	3,010	-	8,991	85,683
	Total	6,300/10%	-	18819/30%	100,000
Model-70%-SL	Cancer	23,035	-	9,828	14,317
	Non-cancer	21,065	-	8,991	85,683
	Total	44,100/70% ^e	-	18819/30%	100,000

a-e: Because the number of patches from each WSI is not the same, the number of patches estimated based on the proportion of extraction is approximate.

f: the labels of the patches are ignored.

R2-Q7:

Figure 2: Please, convert y label to English

Thanks for your suggestion! We now have added the legend AUC to y label in Figure 2.

(a) Dataset-PATT (testing set)

(b) Dataset-PAT

Figure 2. AUC distribution of five models at patch level on Dataset-PATT and Dataset-PAT. The horizontal bar in the box indicates the median, while the cross indicates the mean.

R2-Q8:

"Human-AI competition"

Please describe under which circumstances has the pathologists worked in method (were there time constraints, what other data did they have compared to those they used in "real" clinical

case? [only the images or the patient history, etc... as well...])

Thanks for the comment! Pathologists use these pathological images to diagnose these patients, there is no time limit, and no other clinical data were used. We added this clarification to the revision.

Please see revised Human-AI CRC competition section:

We recruited six pathologists with 1-18 years of independent experience (Supplementary Table 4). They independently reviewed 1,634 WSIs from 10 data centers (Dataset-HAC, Figure 4) with no time limit and diagnosed the cancer solely based on WSIs (i.e., no other clinical data were used).

R2-Q9:

For the AI, are these per patch, per slide or per patient AUCs?

For Human-AI competition, these are per patient AUCs. Thanks!

R2-Q10:

(optional suggestion) You have established before the 10%-SL model performs less well. I would remove it from this graph (fig 4) so you can better adjust the limits of the y-axis.

Thanks for the advice! We removed the model-10%-SL from Fig 4 per your suggestion, as shown in the following.

(a)

(b)

(c)

Figure 4. AUC comparison of in the Human-AI contest using Dataset-HAC, which consists XH-dataset-HAC, PCH, TXH, HPH, ACL, FUS, GPH, SWH, AMU and SYU. Colored lines indicate the AUCs achieved by two models and six pathologists (A-F). The F pathologist didn't attend the competition of SYU, AMU and SWH dataset.

R2-Q11:

the legend says “using Dataset-HAC.” But you show for 9 other datasets as well, no?

Thanks for the question! You are right that we did show the 9 other datasets. The Dataset-HAC is a large dataset used for human-AI competition, and the whole dataset consists of 10 datasets from 10 centers: XH-dataset-HAC, PCH, TXH, HPH, ACL, FUS, GPH, SWH, AMU, SYU.

The legend is now revised as follows.

Figure 4. AUC comparison of in the Human-AI contest using Dataset-HAC, which consists XH-dataset-HAC, PCH, TXH, HPH, ACL, FUS, GPH, SWH, AMU and SYU. Colored lines indicate the AUCs achieved by THE two models and six pathologists (A-F). The F pathologist didn't attend the competition of SYU, AMU and SWH dataset.

R2-Q12:

What is the difference between the 6 pathologists here and those used to select the slides and

their true label? What is the probability that some slides were mis-labelled? When the 6 pathologists made mistakes, what proportions of those mistakes were on the same slides and how did the AI perform for those particular slides misclassified by the pathologists? Do the AI and the pathologist tend to make mistake on the same slides?

Thanks for the comment! Five pathologists (not 6 pathologists) (considered as top experts in the field) reviewed and decided on the true labels, and these 5 pathologists all are well-recognized experts with each having more than 15 years of clinical experience, and they worked together to ensure the agreement, correctness and consistency of the labels as true labels. Moreover, all positive labels had been confirmed by radical resection procedure, thus, it would be nearly impossible that these positive slides were mis-labelled.

The experience of the 6 independent pathologists involved in the contests ranges from a few years to more than ten years. The consistency between 6 pathologist and AI is very good, for example, average Kappa of the model-70%-SL and 6 pathologists was up to 0.896, as shown in our previous work (Wang Kuan-Song, BMC Medicine 2021 [19]). However, we compared the misdiagnosis of the 6 pathologists and found that there was no case where multiple pathologists diagnosed the same slide incorrectly. For slides with incorrect diagnosis by human experts, 91% were incorrectly diagnosed by only 1 (or only a limited few times by 2) pathologists. Not many common mistakes were made by the 6 independent pathologists. There is no obvious consistency between the slides incorrectly diagnosed by AI and slides incorrectly diagnosed by pathological experts.

R2-Q13:

It looks like the pathologists had to decide only whether tumor cells were present or not, right? What was the source of mistakes? Small tumors that were missed, or other cases?

Thanks for the comment! Yes, the existence of tumor cells is an important basis for diagnosis, and other basis includes the tissue structure of the tumor. In practice, there is no room for error due to missing cancer cells by pathologists. The reason for the potential error may be that the pathologist is not very sure whether these cells are tumor cells or normal cells.

R2-Q14:

I appreciate the comparison in Supp Material F (though the section should be referenced in the main text). However, the literature on the more general subject of semi or un-supervised approach for annotation of WSI in general is poorly represented in the paper. Please consider discussing this approach versus other existing approaches that were published to achieve similar goals (for example, Quiros A.C., Murray-Smith R., Yuan K. Learning a low dimensional manifold of real cancer tissue with PathologyGAN. 2020 and others)

Thanks for the comment! We have moved the related work in Supp Material F to Results section of the main text (in subsection of “Comparison with related research”).

Yes, there are some other approaches of similar goals for learning on the small number of labeled data. For example, in the transfer learning, the network is firstly trained in a big dataset of source

domain, and then it is trained in a small number of labeled medical images. The GAN generates large amount of data by learning the style from a limited dataset. Those approaches may improve the cancer recognition, but it is noting that they only used limited labeled datasets, and large number of unlabeled data do appear in medical datasets. It is difficult to show all possible features of the disease with a limited sample. Therefore, a semi-supervised learning method that can simultaneously use unlabeled data seems to be more suitable for medical data.

We have added the discussion in Introduction section.

There are some other approaches for learning on the small amount of labeled data. For example, in transfer learning, the network is firstly trained in a big dataset of source domain, and then trained in labeled medical images. However, the number of labeled images needed is still quite large [22, 23]. The generative adversarial networks (GAN) can generate large amount of data by learning the style from a limited dataset [24, 25]. These approaches may improve accuracy, but they only used limited labeled datasets, and large amounts of unlabeled data do appear in medical domains and clinical settings. Moreover, it would be difficult for GAN to simulate all possible features of the disease based on limited samples.

R2-Q15:

Very important: there is no code available to assess the reproducibility of the work. The code should be made available to the reviewer with a well detailed README fle. If, for confidentiality reasons, the code cannot be shared with the reviewers, then, at least the README file should be shared to ensure that all the details required are there to check future use of the code is possible.

Thanks for the comment! We have shared the code and data in GitHub with the link: www.github.com/csu-bme/pathology_SSL, the readme file is also included in this link.

R2-Q16:

I would be interested in seeing how long it takes to train the network for the different types of models. When do they converge and how was the convergence epoch determined?

Thanks for the comment! For CRC semi-supervised learning, training epochs takes about 10 hours. The proposed network was implemented using TensorFlow framework (Version 1.15.0) [29] and trained on a server consisting of two graphics processing units (GPU) of Tesla V100 32GB, NVIDIA company.

The training period or epoch is one of the hyperparameters. For each model, we adopted a general selection strategy of epoch, one-tenth of the training set was taken out as the validation set, and the training epoch was selected by the best result on the validation set. In order to avoid overfitting, when the accuracy no longer improves on the validation set, the early stopping strategy is adopted. The time to train a network is related to the dataset size, because the more data, the more epochs are needed.

**** Comments on Supp Method A:****R2-Q17:**

At the end of “Digitization and annotation” section of Supp Method A, specify more clearly what happens to the other patches. For example, “In model-5%-SSL [...] small proportions (i.e., 5% and 10%) of total patches (62,919) and masked label information for the remaining patches”. The remaining patches mean 95 and 90% respectively, right? What do you mean by “masked label information”? In table 3, it looks like 30% was used for test and 65% un-used. Is that correct? I don’t see how it fits with what’s said above.

Thanks for the comment! We checked these statements and tried to revise Table 3 for it to be more clear. We first used 30% of all patches as a test set. Then the remaining patches such as 5% or 10% of all patches were used as the semi-supervised training set, and finally the existing labels of the remaining 65% or 60% patches were masked as unlabeled, as illustrated in Table 3 as in the following.

Model	Class	Dataset-PATT			
		Training set		Testing set	Dataset-PAT
		Labeled	unlabeled ^f		
Model-5%-SSL	Cancer	1,645	21,390	9,828	14,317
	Non-cancer	1,505	19,560	8,991	85,683
	Total	3,150/5% ^a	40,950/65% ^b	18819/30% ^c	100,000
Model-10%-SSL	Cancer	3,290	19,745	9,828	14,317
	Non-cancer	3,010	18,055	8,991	85,683
	Total	6,300/10%	37,800/60% ^d	18819/30%	100,000
Model-5%-SL	Cancer	1,645	-	9,828	14,317
	Non-cancer	1,505	-	8,991	85,683
	Total	3,150/5%	-	18819/30%	100,000
Model-10%-SL	Cancer	3,290	-	9,828	14,317
	Non-cancer	3,010	-	8,991	85,683
	Total	6,300/10%	-	18819/30%	100,000
Model-70%-SL	Cancer	23,035	-	9,828	14,317
	Non-cancer	21,065	-	8,991	85,683
	Total	44,100/70% ^e	-	18819/30%	100,000

a-e: Because the number of patches from each WSI is not the same, the number of patches estimated based on the proportion of extraction is approximate.

f: the labels of the patches are ignored.

R2-Q18:

For the other model, “In model-5%-SL, model-10%-SL and model-70%-SL, we used 65% supervised training with 5%, 10%, 70% of the total 62,919 patches”, what happens to the remaining 95, 90 and 30%? Are they used as PATT test as in Table 3?

Thanks for the question for us to clarify! For model-5%-SL, 5% and 30% patches of 62,919 were used for training and testing set respectively, and the remaining 65% patches were not used. Similarly, for model-10%-SL, the 10% and 30% patches were used for training and testing set respectively, and the remaining 60% patches were not used. For model-70%-SL, the 70% and 30% patches were used for training and testing set respectively.

We revised Table 3 for better explanation accordingly, please see R2-Q17.

R2-Q19:

“Because the WSI is very large (>50,000 pixels), the patch-level models were trained to recognize cancerous probability, and all the patch-level results on a WSI were combined to infer the cancerous probability of the WSI. The flow chart is shown in Figure 1.”How were they combined exactly?

Thanks for the comment! In theory, if the patch diagnosis is 100% accurate, as long as a patch is recognized as cancerous, the WSI should be considered as the cancerous. But the accuracy of the patch-level model is not 100%, so the false positive errors of WSIs will be accumulated if we solely use the diagnosis on individual patches. We proposed a novel clustering-based strategy earlier (Wang Kuan-Song, BMC Medicine 2021 [19]) to control the false positive errors. In this strategy, once three consecutive patches are predicted to be cancerous, the probability of false positive errors at the WSI level will be reduced to one in ten thousand (Please see Supplementary B). Experiments have proved that if 4 consecutive patches are identified as cancerous, the maximum AUC of WSI can be obtained (Wang Kuan-Song, BMC Medicine 2021 [19]).

We clarified the issue in the Supplementary A as in the following.

Clustered-based WSI inference

Because the accuracy of patch-level models cannot be 100%, there were serious false positives in WSI predictions if any patch in the WSI was identified as positive (cancer) and used as a criterion for predicting the WSI cancerous status. Intuitively, because the tissues in WSI were continuous, the area with cancer should be distributed continuously and included several continuous positive patches. This intuition had been used to effectively control the false-positive of functional magnetic resonance images [4]. We designed a simple clustering-based inference method. If some continuous patches were identified as having cancer by patch-level model, the cancer may indeed exist on WSI. The cluster size of four patches was expected to best control the false-positive rate as shown in our early study [2], that is, the condition of continuously identifying 4 patches with cancer on WSI was used as the basis for determining the existence of cancer in WSI. For statistical analysis on patient-level prediction, please refer to Supplementary B.

Patient-level diagnosis

Clinically, multiple WSIs may be obtained for one patient. The inference on patient level was based on positive sensitivity, that is, if all WSIs from the same patient were identified as negative (no cancer), then the patient was negative, otherwise the patient was positive.

R2-Q20:

“We tested the known CNNs, such as VGG16, ResNET V1 and V2, Inception V1-V4,

Mobilenet, etc., and found that InceptionV3 [16] achieved most consistent results on many datasets. Therefore, we used Inception V3 as the baseline model.”

- Please, do not use “etc”. Be specific which ones were used

- You state “InceptionV3 [16] achieved most consistent results on many datasets.” Since you’ve done those tests, please show the results in a (supp?) figure to support that observation

Thanks for the comment! We removed the “etc”, since we already listed all the methods tested. Our previous paper (Wang Kuan-Song, BMC Medicine 2021 [19]) trained the networks for CRC diagnosis by supervised learning, and listed the accuracy of some networks such as Inception V1, V3 and V4 on the CRC datasets. Among them, Inception V3 showed the best accuracy. We cited our previous work as reference to support this observation. Because the baseline model is based on supervised learning on the CRC datasets, the semi-supervised models here also choose Inception V3 in order to be compared with the previous supervised learning study in paper (Wang Kuan-Song, BMC Medicine 2021 [19]).

We revised the Patch-level SSL and SL subsection in Supplementary A

Our previous work tested some known CNNs, such as VGG16, ResNET V1 and V2, Inception V1-V4, and found that Inception V3 [1] achieved most consistent results on the CRC datasets [2]. Therefore, we used Inception V3 as the baseline model of SL.

R2-Q21:

“The patch-level models included supervised and semi-supervised versions. The input patch size was scaled to 299×299”

Before, you said you extracted 300x300 pixel patches. Was it scaled from 300x300 to 299x299? This sounds weird. Why not extract the patches directly at 299x299?

Thanks for the comment! The patch size we labeled is 300*300, which can be easily transformed to the required input size of most CNN architectures. For example, the input accepted by Inception V3 is 299*299, and the input accepted by VGG16 is 224*224. In order to input these patches to Inception V3, we used the bilinear interpolation method [21] to resize the patches from 300*300 to 299*299.

We added the information in Patch-level models section of Supplementary A as follows.

The patch size we labeled is 300×300 as it can be easily transformed to the required input size of most CNN architectures. We used the bilinear interpolation method [21] to scale the patch size to 299×299, which is the default input size of Inception V3.

[21] Han D. Comparison of commonly used image interpolation methods. Conference of the 2nd International Conference on Computer Science and Electronics Engineering. 2013;1556-1559.

R2-Q22:

“The semi-supervised version was based on the mean teacher method [14], where two Inception V3 were trained, one as student and the other as teacher.” Please reference Supp

material C. Also, the paper itself would benefit from a clearer description of this architecture.

Thanks, we have referenced the supplementary material C (Supplementary A in this current revised version) in main text, and described this architecture more in supplementary A, including the flow chart, loss function, and training methods. We listed the revised paragraphs below.

We added the reference in methods section of manuscript:

For SSL, we applied an SSL strategy called mean teacher [20], where two Inception V3 networks, teacher network and student network were trained.

At the WSI and patient levels, we applied a cluster-based and positive sensitivity strategy to achieve CRC diagnosis for patients as we did recently [19] (Figure 1). To save space, the details of the CRC diagnosis methodology were relegated to Supplementary A.

R2-Q23:

“The Inception V3 adopted the pre-trained model on ImageNet database, and was deeply fine-tuned.”

What do you mean? Did you do transfer learning, or do you mean something else? Also, what fine-tuning was done exactly?

Thanks for the comment! Yes, this is transfer learning. We initialized the Inception V3 with a pre-trained model on ImageNet database, and then trained it on our pathological images. During training, the weights in all layers of inception V3 were updated, which is referring to as deeply fine-tuned.

We have clarified it in network training at patch level subsection of Supplementary A as in the following:

The Inception V3 was initialized with a pre-trained model on ImageNet database, and then trained on the pathological images. During training, the weights in all layers of inception V3 were updated.

R2-Q24:

“The training cycle was 100 epochs, each epoch included 100 steps.”

And what was the batch size in each step? Since you say “because of the imbalance between the labeled and unlabeled data, we maintained the same proportion of labeled and unlabeled patches in each mini-batch.”, I guess it means some labelled patches were used several times and unlabeled patches once. Is that the case? In this particular case then, how is “1 epoch” defined exactly then?

Thanks for the comment! In the experiments, the batch size is set to 128. Because there are more unlabeled patches than labeled patches, the labeled patches have indeed been used more times than unlabeled patches. Here, an epoch is defined as the patches of $100 \times \text{batch_size}$.

We revised the paragraph 3 of network training and clarified it in Supplementary A as follows.

In the SSL, because of the imbalance between the labeled and unlabeled data, we maintained the same proportion of labeled and unlabeled patches in each mini-batch of 128 patches. The optimizer was Adam. The training period was 500 epochs, and each epoch included 100 steps. If the

accuracy on validation set can't be improved for 80 consecutive epochs, the early stopping [22] was applied. In order to prevent the training from ending prematurely, 50 epochs for pre-training were executed before the early stopping. L2 decay was used and the decay coefficient was set to 0.0001. The teacher network was initialized with the student network. The student network would update the weights in each step, but the teacher network used exponential moving average to update the weights after one epoch ended. The smoothing coefficient was set to 0.95.

R2-Q25:

“Clustered-based WSI inference” section: what if some patches are at the edge of two regions, one made of non-cancerous cells, the other of cancerous cells?

Thanks for the comment! When the human tissue is magnified by 20X, one patch may contain many cells. As long as there is one cancerous cell in the patch, the patch is considered as a cancerous patch. When the patch is at the edge of cancerous region and benign region, there is at least one cancerous cell or none in the patch is the rule for patch-level predication. Next, all patches that are positive or cancerous are analyzed together, and if they can form clusters with size ≥ 4 , the WSI are considered to be positive.

Therefore, no matter where these patches are, as long as there are cancer cells, they are considered positive. If 4 positive patches can form a cluster, the WSI is considered positive.

R2-Q26:

Supp table 4: for the “other cancers” section, please specify for each of the which cancer

Thanks, we now specified the cancers as lung, ductal carcinoma, breast, which is added in the Supplementary Table 10 in the revision (Supplementary Table 4 in the last reviewed version).

**** Other minor comments:**

R2-Q27:

Fig 1 would benefit from more details on the legend (for example, below the models on the right is written things like “5%/10%-SL” - Please explain in the legend that it is the model name). Since the Fig 1 is referred before explaining the differences or the associated tables, legend should be more complete.

Thanks, we have explained the legend before the Fig1 is referred. We first described the models, and then refer to Table 1 and then Figure 1. Please see the beginning of the Result section.

In paragraph 1 in the Methods section:

We trained and tested our method utilizing CRC datasets from multiple centers (Table 1).

We used SSL, SL to represent semi-supervised and supervised learning methods, and a numerical number to represent labels used in training set, which accounts for the percentage of 62,919 patches in Dataset-PATT (Table 2). Five versions of patch-level model, Model-n%-SSL or Model-n%-SL were obtained based on different training sets and learning method (Table 3, Figure 1 (a)) and then tested (Figure 1 (b)). For SL, Inception V3 [36] was used for patch-level model. For

SSL, we applied a SSL strategy called mean teacher method [26], where two Inception V3 networks, teacher network and student network were trained.

We also provided a revised Figure 1 as shown in the following to clarify training and testing of the models.

Figure 1. The flow chart of the CRC study. (a) SSL and SL are performed on different labeled and unlabeled patches from Dataset-PATT. (b) The patch-level test on 30% of Dataset-PATT and whole dataset of Dataset-PAT. (c) The patient-level test of Dataset-PT and Dataset-HAC. If there is a cluster of four positive patches on WSI, the WSI is positive. A subject with a positive WSI is cancerous.

R2-Q28:

PAT – table 1 and 2: why are the number of subjects/tiles unknown? How were those obtained?

The dataset was downloaded from the NCT-UMM <https://zenodo.org/record/1214456#.XV2cJeg3lhF>, there are a total of 100,000 patches from 86 slides, but no information on the number of cancer and non-cancer subjects/slides was provided. We

clarified it in revised Table 2 as shown in the following.

Table 2. Dataset-PATT and Dataset-PAT

Dataset	Cancer			Non-cancer			Total		
	subjects	slides	patches	subjects	slides	patches	subjects	slides	patches
Dataset-PATT	614	614	30056	228	228	32863	842	842	62919
Dataset-PAT^a	NA	NA	14,317	NA	NA	85,683	NA	86	100,000
Total	>614	>614	44,373	>228	>228	118,546	>842	928	162,919

a: No information (NA) on the number of cancer or non-cancer subjects/slides was provided.

R2-Q29:

Supp Method A: “From the 842 WSIs in Dataset-PATT, two pathologists manually selected some representative patches,”

How was it done? Did they draw selection directly on the WSI from which patches were extracted afterwards (in which case, which software package was used), or were the patches extracted first, and annotated after by the pathologists (in which case, how were the 100 patches max per WSI pre-selected, and why are they ~60,000 patches in the end instead of 842x100 patches?)?

Thanks for the comment! A lab technician randomly retrieved 842 WSIs from pathological library of Xiangya hospital and then these WSIs were scanned using a KF-PRO-005 scanner (KFBIO company, Ningbo City, China) at 20X magnification. Because the shape of the CRC tissue is less diverse than that of non-cancerous tissue, more cancer positive WSIs (n=614) and less cancer negative WSIs (n=228) were selected.

Two pathologists used image browser software provided by KFRIO company (Ningbo City, China) from one WSI to export some regions of interest (ROI) according to the size of WSI. In order to maintain the diversity of cancer cell distribution, 4-10 positive ROIs were extracted from each positive WSI. In order to ensure that the number of positive ROIs and negative ROIs is balanced, 10-25 ROIs are extracted from each negative WSI. One ROI has a size of about 1024*768 pixels, and was split into about 6 non-overlapping patches with 300*300 pixels in order to be adaptable to meet the input size of most neural networks. The two pathologists then manually reviewed the patches, each of which was labeled with either cancer or cancer-free. When the two pathologists reached a consensus on the annotation of patches, which were kept in the Dataset-PATT. Finally, a total of 30,056 patches with cancer and 32,863 patches without cancer were obtained. Because the patches obtained from a considerable part of WSIs are less than 100, the total number is only 62919.

We clarified the issue in the annotation of patches in Dataset-PATT in Supplementary A.

R2-Q30:

Only Supplementary A is referred in the main text, others are not.

Thanks, we add the references to supplementary materials A-D in the text.

R2-Q31:

Only 2 scenarios were tested (5%-SSL and 10%-SSL). I understand it is impossible to test all combination, but can the authors comment (or do a kind of power analysis) on, in general, what would be the optimal percentage of labeled data needed or how to find it in an efficient way? Also, since you are not limited to the number of unlabeled data, would the performance be even more improved if you were to use a much higher number of unlabeled patches?

Thanks for the suggestion! Yes, because deep learning requires a lot of time to train, it is impossible to make the test for all possible ratios of labeled data. Generally speaking, the more labeled data, the better the result, there is no theory to accurately determine the appropriate ratio. Just like the experiments with CRC and lung cancer data, more labeled data improve performance. For example, model-10%-SSL is better than model-5%-SSL.

It would be reasonable to extrapolate that more unlabeled data may improve the results, which is an interesting question. However, we guess the increased amount of unlabeled data may not always improve performance. In the datasets of lymph node cancer, there are 262,144 patches in the training set. We trained two SSL models, one using 2,622 labeled and 129,761 unlabeled patches as the training data set, the other using 2,622 labeled and 259,522 unlabeled patches as the training data set. The testing set included 32,768 patches (16,384 cancer and 16,384 normal). The AUC of first model is 0.945, and the AUC of second model is 0.947. There is no statistical difference. The experiment indicated that more unlabeled patches may not always improve the performance.

training set		testing set	AUC
labeled patches	unlabeled patches		
2,622	129,761	32,768	0.945
2,622	259,522	32,768	0.947

Therefore, the manuscript has demonstrated that the final performance is related to the amount of labeled data such as 5% and 10%, the unlabeled data can improve the performance. However, we guess more unlabeled data may not always further improve performance, for example lymph node experiments above. How to improve the performance by additional unlabeled data would need further be investigated in the future work.

**** As a conclusion:**

R2-Q32:

The paper overall is quite interesting. I think developing architectures or strategies that allow good prediction with few labels is important.

Strengths of the paper:

- **The number of slides used is high and the dataset comes from a very large number of different institutions, supporting therefore well the conclusions of the paper**
- **Comparison with 6 pathologists is performed**
- **The application is important to the field to reduce the number of labeled data needed**

Major weaknesses that would need to be addressed to make the approach more sound/robust:

- many sections would benefit from major clarifications. Also making sure, in the workflow, that dataset, acronyms and data are well explained the very first time they are mentioned in the text would help the flow.

- **Description of the architecture**
- **Code availability (or README file) missing**

We thank the reviewer for the positive summary and constructive comments and suggestions! We have made every effort to improve the quality of our current work (as summarized in the above responses) and hope you will find it helpful and the manuscript improved per your kind advice.

Reviewer #3 (Remarks to the Author):

The proposed work investigates semi-supervised deep learning for the classification of colorectal cancer on pathological images. A large dataset with 13,111 whole slide images was used for the investigation, and the results show that, with a relatively small number of annotated training samples, semi-supervised classification can be comparable to fully-supervised learning with a large number of annotated training samples. These results suggest that semi-supervised learning can reduce the cost of annotations of colorectal cancer and/or allow decent classification performance when the annotations are scarce.

The problem of scarce annotations in medical imaging is in general an important one to address and indeed requires attention. The proposed work could make an impact. However, semi-supervised deep learning has already been explored in many studies for medical imaging (for example, see [1] for a review of relevant studies). Although the application to colorectal cancer is new, the proposed work uses the well-established mean teacher method, which is quite commonly used for medical image classification (see [2] and [3] for example). Thus, the novelty in terms of application is incremental, and the technical contribution is little.

R3-Q1:

As for the experiments, it is a strength that a relatively large number of images from different centers were included. However, considering the limited novelty in terms of application and methodology, I would suggest experiments with a wider range of diseases to increase the impact of the proposed work. Can the authors test three or four different diseases on whole slide images, and then conclude that semi-supervised deep learning is beneficial for general image classification based on whole slide images?

Thanks for the suggestion! In order to illustrate the application of semi-supervised methods in pathological images, we did additional experiments on 2 other types of cancer: lung and lymph node cancers. Because of the lack of a dataset as large as CRC, we only did the assessment of two cancers at the patch level. We revised the manuscript accordingly and listed the revisions below for your reference.

(1) Extended SSL vs SL recognition of lung and lymph node cancer section:

In order to demonstrate the utility of SSL on other pathological images, the experiment of SSL and SL on lung and lymph node were performed as detailed in Supplementary D for Methodology. 15,000 lung images of three classes: adenocarcinoma, squamous cell carcinoma, and benign tissue were obtained from LC25000 dataset (Lung) [38], and the 294,912 lymph node images including tumor and benign tissue were obtained from PatchCamelyon (Pcam) [39]. Similarly, SSL was trained on a small number of labeled images and a large number of unlabeled images (for which labels are known but ignored during training), and compared with SL (Supplementary Tables 6 and 7). The base model of SL is Inception V3, and mean teacher method is also used for SSL, repeatedly trained 8 times as in the main CRC study to assess various performance statistics.

Because the number of classes in lung images is three, the accuracy is used for the evaluation. Lung-5%-SSL (5% labeled and 75% unlabeled) and Lung-20%-SSL (20% labeled, 60% unlabeled)

were better than Lung-5%-SL (5% labeled) and Lung-20%-SL (20% labeled) (Accuracy: 0.960 ± 0.007 vs 0.918 ± 0.026 , P value=0.012; 0.989 ± 0.003 vs 0.961 ± 0.025 , P value=0.011, Figure 5) respectively. There was no difference between Lung-20%-SSL and Lung-80%-SL (80% labeled) (Accuracy: 0.989 ± 0.003 vs 0.993 ± 0.002 , P value=0.093). Pcam-1%-SSL (1% labeled, 99% unlabeled) and Pcam-5%-SSL (5% labeled, 95% unlabeled) are better than Pcam-1%-SL (1% labeled) and Pcam-5%-SL (5% labeled) (AUC: 0.947 ± 0.01 vs 0.912 ± 0.01 , P value=0.012; 0.960 ± 0.002 vs 0.943 ± 0.01 , P value=0.0001, Figure 6) respectively. Pcam-5%-SSL can be compared to Pcam-100%-SL (100% labeled) (AUC: 0.960 ± 0.002 vs 0.961 ± 0.005 , P value=0.94). This extended experiment confirmed the conclusion that when a small number of labeled pathological images was available together with a large number of unlabeled image data, SSL can be compared to SL with massive labels.

(2) In Paragraph 8 in the Discussion section, we added the following:

Based on the extended experiments for cancers of lung and lymph node, we further confirmed the conclusion on CRC that when a small amount of labeled data was used, SSL plus a large amount of unlabeled data, performed better than SL (with the same number of labeled images). SSL performance can be compared to SL with massive labeling, which confirms the conclusion that SSL may reduce need for the amount of annotation data on pathological images.

R3-Q2:

The authors claim that "how to translate the patch-level prediction to WSI and patient level diagnosis is not trivial". Please elaborate why.

Thanks for the comment! Because WSI is too large, the patient level model is always developed based on the patch level. So far, we have not been able to get a perfect patch-level model, which has 100% sensitivity and 100% specificity. If the sensitivity of the patch-level model is not too low, the sensitivity of WSI/patient may be increased because only one positive patch of large number of patches on WSI is found, and the WSI can be consider as positive. However, if its sensitivity is too low, all positive patches may not be found. If we make the sensitivity of patch-level model increased to improve the sensitivity of WSI/patient, its specificity will decrease accordingly. The patch-level model with low specificity will consider some negative patches as positive, so false positive errors of WSIs will be accumulated with multiple patches of WSIs. Therefore, how to get a perfect patient-level inference based on an imperfect patch-level model is not an easy task and requires studies.

In Paragraph 5 of Introduction section, we revised as below:

Because we and other groups have not been able to develop perfect patch-level models, the errors at patch-level may be easily magnified on WSI level diagnosis. For example, even though the imperfect patch-level model may yield reasonable prediction on positive (cancerous) WSIs, it also may yield high false positive errors on the negative (non-cancer) WSIs, because the false positive errors at patch level will accumulate due to the testing of multiple patches in WSI. However, the patient-level diagnosis is required in clinical applications of any AI system for cancer diagnosis.

R3-Q3:

I am also somewhat confused about the generation of patch-level annotations. While I believe the patient-level annotation is determined by whether cancer exists in the image of the patient, for the patches, do we need experts to look at each patch and determine if cancer exists in the patch too? If that is the case, the annotation can be quite laborious. Please correct me if I understand it incorrectly and clarify.

Thanks for the comment! Yes, for artificial intelligence development, the pathological images need to be labeled for each patch, since the machine learning tools cannot take the whole data from WSI and can only take smaller amount of data (in patches) chopped from WSI. This is indeed a very time-consuming and labor-intensive process. Therefore, how to reduce the number of annotations needed for artificial intelligence analyses via deep learning is a problem worthy of research, highlighting the significance of our current work.

R3-Q4:

In Table 4, row 6 is a repeat of row 4. Can there be a more efficient way of presentation without such repetitions?

Thanks, we repeated the row 6 to compare Model-10%-SSL with the two supervised models. In the modified version, we removed the Table 4, because Figure 2 (shown below) repeats the contents of Table 4.

(a) Dataset-PATT (testing set)

(b) Dataset-PAT

Figure 2. AUC distribution of five models at patch level on Dataset-PATT and Dataset-PAT. The horizontal bar in the box indicates the median, while the cross indicates the mean. Circles represent data points.

R3-Q5:

There are some grammatical errors. For example, in line 192, "models currently relies" -> "models currently rely". Please double check thoroughly.

References

- [1] Cheplygina et al., Not-so-supervised: A survey of semi-supervised, multi-instance, and transfer learning in medical image analysis, *Medical Image Analysis*, 2019.
- [2] Su et al., Local and Global Consistency Regularized Mean Teacher for Semi-supervised Nuclei Classification, *International Conference on Medical Image Computing and Computer-Assisted Intervention*, 2019.
- [3] Liu et al., Semi-supervised Medical Image Classification with Relation-driven Self-ensembling Model, *IEEE Transactions on Medical Imaging*, 2020.

Many thanks for your constructive comments, helpful suggestions and references! We have carefully tried to proofread and revise the manuscript. We added the above references in Introduction from [20] to [22].

Reviewers' Comments:

Reviewer #1:

Remarks to the Author:

Thank you very much for answering most of my queries. Most scientific queries have been resolved. However, the figures still look very unprofessional and most of the article will require heavy copyediting to meet the standards we usually see in Nature Communications.

Reviewer #2:

Remarks to the Author:

The authors addressed the comments and the paper is much clearer than initially.

At the light of the clarified method, I have one more question:

Do the unlabelled patches potentially belong to the same slides / patients than the labelled patches? If so, then they will look more similar than if they were not, and we can argue that unlabeled patches coming from same patients where slides were already labelled are more beneficial than non-labelled slides coming from new patients from new institutions? If images from a same patient can indeed be found in both "labelled" and "unlabelled" sets of the training dataset, then this must be clearly specified and discussed, because the source of the unlabelled data might then matter.

Two minor comments:

1- Line 189-190: "The performance of Model-10%-SSL had no significant difference with that of Model-70%-SL (AUC: 0.98 ± 0.01 vs. 0.987 ± 0.01 , P value=0.134). "

What this for the PATT or the PAT dataset? Please report both and comment if both are statistically similar or not.

2- What is the code / reference for the heatmaps generated in Supp Fig 2?

Reviewer #3:

Remarks to the Author:

The authors have made good efforts to address my concerns. I have some additional questions regarding the new experimental results as well as the annotation strategy.

The authors state that "because the number of classes in lung images is three, the accuracy is used for the evaluation." This can be problematic when classes are imbalanced. The classifier can simply label the samples as the majority class to achieve good accuracy. The authors could report class-wise AUC instead, or class-wise precision and recall.

There are also some recent works [1,2] on the annotation strategy for WSI, where only a fraction of positive samples are annotated, and the other samples are unlabeled. This strategy may further reduce the annotation effort, as the experts do not need to carefully ensure that there are no positive instances in the negative image. Then, network training can be performed with loss calibration or positive-unlabeled learning [3]. Although these works are originally developed for cell detection, they can be adapted for WSI classification. The authors could consider discussing the possibility of such annotation in future work.

References

[1] Li et al., A Novel Loss Calibration Strategy for Object Detection Networks Training on Sparsely Annotated Pathological Datasets, MICCAI 2020.

[2] Zhao et al., Positive-unlabeled Learning for Cell Detection in Histopathology Images with Incomplete Annotations, arXiv:2106.15918

[3] Fabien et al., Learning from positive and unlabeled examples, Algorithmic Learning Theory, 2000.

Response to reviewers' comments

Reviewer #1:

Thank you very much for answering most of my queries. Most scientific queries have been resolved. However, the figures still look very unprofessional and most of the article will require heavy copyediting to meet the standards we usually see in Nature Communications.

R1-Q1:

Thank you very much for pointing this out. We have revised Figures 2 to 6 significantly to meet the standards required by Nature Communications.

Figure 2. AUC distribution of five models at patch level on testing set of Dataset-PATT and Dataset-PAT. The horizontal bar in the box indicates the median, while the cross indicates the mean. The circles represent data points. * indicates significance difference, and ** indicates no significance difference.

(a) Model-10%-SSL

(b) Model-10%-SL

(c) Model-70%-SL

Figure 3. Patient-level comparison of three models on twelve independent datasets from Dataset-PT. Left: Radar maps illustrating the sensitivity, specificity, and AUC. Right: Boxplots showing the distribution of sensitivity, specificity, accuracy, and AUC in these datasets. The horizontal bar in the box indicates the median, while the cross indicates the mean. The circles indicate the data points, which were listed in Supplementary Table 3.

Figure 4. AUC comparison of model-10%-SSL(SSL), model-70%-SL(SL) and six pathologists (A-F) in the Human-AI contest using Dataset-HAC, which consists XH-dataset-HAC, PCH, TXH, HPH, ACL, FUS, GPH, SWH, AMU and SYU. Blue lines indicate the AUCs achieved by model-10%-SSL. The F pathologist didn't attend the competition of SYU, AMU and SWH dataset.

Figure 5. Accuracy distribution of five models on LC25000 dataset. The horizontal bar in the box indicates the median, while the cross indicates the mean. The circles indicate the data points.

Figure 6. AUC distribution of five models on PatchCamelyon dataset. The horizontal bar in the box indicates the median, the cross indicates the mean, while circles indicate the data points.

Reviewer #2:

The authors addressed the comments and the paper is much clearer than initially.

At the light of the clarified method, I have one more question:

Do the unlabelled patches potentially belong to the same slides / patients than the labelled patches? If so, then they will look more similar than if they were not, and we can argue that unlabeled patches coming from same patients where slides were already labelled are more beneficial than non-labelled slides coming from new patients from new institutions? If images from a same patient can indeed be found in both “labelled” and “unlabelled” sets of the training dataset, then this must be clearly specified and discussed, because the source of the unlabelled data might then matter.

Two minor comments:

1- Line 189-190: “The performance of Model-10%-SSL had no significant difference with that of Model-70%-SL (AUC: 0.98 ± 0.01 vs. 0.987 ± 0.01 , P value=0.134). “

What this for the PATT or the PAT dataset? Please report both and comment if both are statistically similar or not.

2- What is the code / reference for the heatmaps generated in Supp Fig 2?

R2-Q1:

Do the unlabelled patches potentially belong to the same slides / patients than the labelled patches?

We thank the reviewer for pointing this out. Firstly, the Dataset-PATT was randomly divided into training set and testing set, and the patches from the same subject/WSI would not be in different sets, to ensure independence of the different data sets. Meanwhile, the patches from 70% of 842 subjects/WSIs were used as the training set, while the remaining patches from 30% subjects/WSIs were used as the testing set.

The unlabeled patches and the labeled patches in the training set may belong to the same slides / patients. Please refer to the response to R2-Q2.

R2-Q2:

If so, then they will look more similar than if they were not, and we can argue that unlabeled patches coming from same patients where slides were already labelled are more beneficial than non-labelled slides coming from new patients from new institutions? If images from a same patient can indeed be found in both “labelled” and “unlabelled” sets of the training dataset, then this must be clearly specified and discussed, because the source of the unlabelled data might then matter.

Because the WSI is too large to input to the neural network, the current method is first based on patch-level recognition. The usual practice is to scan dozens to hundreds of WSI and prepare a

patch-level dataset. Pathologists select representative and non-overlapping patches images from these WSIs for annotation. There are about tens to hundreds of patches selected in each WSI. For each patch, there are generally several pathologists who independently label them, and only those who reach an agreement can be included in the data set. This annotation is very detailed, and only the patch images that are completely correct, including the typical structure of the disease, can be used for patch-level model training and testing. The number of these patches is as many as tens of thousands, so the annotation takes a lot of time.

When the number of WSIs (70% of 842 WSIs) in the training set is known, there are two ways to reduce the labeling effort on the patches from these WSIs. First, we can reduce the number of WSIs with labelled patches, but the number of labeled patches per WSI does not decrease; Second, we do not reduce the number of WSIs, but reduce the number of labeled patches marked on each WSI.

We agree that the patch images of WSI from the same patient have a certain similarity, which is determined by the characteristics of pathological images. It looks reasonable to label patches from some WSIs and not to label patches from other WSIs. However, there are challenges and disadvantages in such implementation. First, the image coloring process determines that the background color of the same WSI is similar, but different from other WSIs. Second, if the labeled patches are from limited number of WSIs, some typical cell structure of the disease may not be included, which may greatly reduce the generalizability of the trained model. More importantly, SSL theoretically assumes that data points, both labelled and unlabeled, are smooth [23]. In other words, the labels of unlabeled patches are potentially determined by neighboring labeled patches in the feature space. If the labeled patches and the unlabeled patches come from different WSIs, the distance of labeled and unlabeled patch will unavoidably include the differences of colors and tissue structures among WSIs included in the training sets, thereby the smoothness assumption among data points is violated.

Our semi-supervised method used the second way. It does not reduce the number of labeled WSIs, but it reduces the number of patches that need to be labeled for each WSI. Our experiments on the three types of studied cancer have proved that for each WSI, there is no need to label as many patches as before, but only need to label a small proportion (~5%-10%) of what is required for supervised learning in the past, and it is possible to achieve similar results. In future semi-supervised labeling, we suggest that patient experts select a small number of patches from each WSI, such as a few patches and label them, and other unselected patches from the same sets of WSIs are used as unlabeled patches for semi-supervised learning, which will greatly reduce the needed number of labeled patches.

In summary, we recommend that in future pathological image labeling, we can reduce the number of labeled patches for each WSI, thereby reducing the total number of labels required. Second, unlabeled patches from the same sets of WSIs are helpful for training. Performing semi-supervised learning together with these unlabeled and labeled patches from same slide/patient can achieve similar results to the supervised learning of labeling a large number of patches in the same WSI. Third, semi-supervised learning is more efficient because it reduces the total number of labeled

patches.

We added the discussion of annotation details in the Methodology (Patch-level SSL and SL models Subsection in Supplementary A).

The Dataset-PATT was randomly divided into training set and testing set according to the proportions showed in Table 3, and the patches from the same subject/WSI would not be in different sets, to ensure independence of the different data sets. Meanwhile, the patches from 70% of 842 subjects/WSIs were used as the training set, while the remaining patches from 30% subjects/WSIs were used as the testing set.

When the number of WSIs (70% of 842 WSIs) in the training set is known, there are two ways to reduce the labeling effort on the patches from these WSIs. The first method is that the patches from some WSIs are labeled, while the patches from other WSIs are unlabeled. However, there are some differences between WSIs such as staining, disease subtypes. SSL theoretically assumes that data points, both labelled and unlabeled, are smooth [23]. In other words, the labels of unlabeled patches are potentially determined by neighboring labeled patches in the feature space. If the labeled patches and the unlabeled patches come from different WSIs, the distance of labeled and unlabeled patch will unavoidably include the differences of colors and tissue structures among WSIs included in the training sets, thereby the smoothness assumption among data points is violated.

By contrast, because the labeled patches and unlabeled patches from the same WSI are similar and will not be affected by differences between WSIs. The smoothness assumption of SSL can be better met. Therefore, in order to extract n% (5%, 10%, 70%) of total patches (62,919) as the labeled patches for training, we used another method that is randomly selecting and labeling the n%/70% of all the patches from each WSI in the training set, and the remaining patches of the WSI are not labeled (labels are masked).

[23] Olivier Chapelle, Bernhard Scholkopf, Alexander Zien, Semi-Supervised Learning, MIT Press, 2006.

R2-Q3:

Line 189-190: “The performance of Model-10%-SSL had no significant difference with that of Model-70%-SL (AUC: 0.98 ± 0.01 vs. 0.987 ± 0.01 , P value=0.134). “

What this for the PATT or the PAT dataset? Please report both and comment if both are statistically similar or not.

Thank you very much for pointing this out! The AUC of Model-10%-SSL and Model-70%-SL on both PATT and the PAT dataset have been compared in Fig.2. There is also no significant difference between the results of Model-10%-SSL and Model-70%-SL on the PATT or the PAT

dataset. We added the reports on PATT and PAT in the SSL vs SL CRC recognition subsections of main text.

We also reported these averages and standard deviations of performance, and keep them to the third decimal place.

Result – SSL vs SL CRC recognition at patch level subsections.

The AUC distribution on Dataset-PATT and Dataset-PAT were shown in Figure 2. Model-5%-SSL was superior to Model-5%-SL (Average AUC and standard deviation in Dataset-PATT: 0.906 ± 0.064 vs. 0.789 ± 0.016 , P value=0.017; Dataset-PAT: 0.948 ± 0.041 vs. 0.898 ± 0.029 , P values=0.017; Both Dataset-PATT and Dataset-PAT: 0.927 ± 0.058 vs. 0.843 ± 0.059 , P value=0.002, Wilcoxon signed rank test). Model-10%-SSL was also significantly better than Model-10%-SL (AUC in Dataset-PATT: 0.990 ± 0.009 vs. 0.944 ± 0.032 , P value=0.012; Dataset-PAT: 0.970 ± 0.012 vs. 0.908 ± 0.024 , P values=0.012; Both: 0.980 ± 0.014 vs. 0.926 ± 0.034 , P value=0.0004). These results indicated that when approximately 3,150 (5%) or 6,300 (10%) patches were labeled, the SSL was always better than SL.

The performance of Model-10%-SSL had no significant difference with that of Model-70%-SL (AUC in Dataset-PATT: 0.990 ± 0.009 vs. 0.994 ± 0.004 , P value=0.327; Dataset-PAT: 0.970 ± 0.012 vs. 0.979 ± 0.005 , P values=0.263; Both: 0.980 ± 0.014 vs. 0.987 ± 0.008 , P value=0.134). This observation indicated that there was no significant difference between the SSL method (6,300 labeled, 37,800 unlabeled) and the SL (44,100 labeled). These results indicated that when approximately 3,150 (5%) or 6,300 (10%) patches were labeled, the SSL was always better than SL.

R2-Q4:

What is the code / reference for the heatmaps generated in Supp Fig 2?

Thank you very much for your comment! We have added the reference [24] to which we have referred to generate the heatmaps in Supp Fig 2.

[24] Selvaraju R R., Cogswell M., Das A., et al. Grad-CAM: visual explanations from deep networks via gradient-based localization. IEEE International Conference on Computer Vision. (2017).

Reviewer #3:

The authors have made good efforts to address my concerns. I have some additional questions regarding the new experimental results as well as the annotation strategy.

The authors state that "because the number of classes in lung images is three, the accuracy is used for the evaluation." This can be problematic when classes are imbalanced. The classifier can simply label the samples as the majority class to achieve good accuracy. The authors could report class-wise AUC instead, or class-wise precision and recall.

There are also some recent works [1,2] on the annotation strategy for WSI, where only a fraction of positive samples are annotated, and the other samples are unlabeled. This strategy may further reduce the annotation effort, as the experts do not need to carefully ensure that there are no positive instances in the negative image. Then, network training can be performed with loss calibration or positive-unlabeled learning [3]. Although these works are originally developed for cell detection, they can be adapted for WSI classification. The authors could consider discussing the possibility of such annotation in future work.

References

- [1] Li et al., A Novel Loss Calibration Strategy for Object Detection Networks Training on Sparsely Annotated Pathological Datasets, MICCAI 2020.
- [2] Zhao et al., Positive-unlabeled Learning for Cell Detection in Histopathology Images with Incomplete Annotations, arXiv:2106.15918
- [3] Fabien et al., Learning from positive and unlabeled examples, Algorithmic Learning Theory, 2000.

R3-Q1 :

The authors state that "because the number of classes in lung images is three, the accuracy is used for the evaluation." This can be problematic when classes are imbalanced. The classifier can simply label the samples as the majority class to achieve good accuracy. The authors could report class-wise AUC instead, or class-wise precision and recall.

Thank you very much for this valuable question. Because the number of images of the three classes in the lung cancer experiment is balanced (5000 images per class), we used accuracy to report the results. As you said, "when classes are imbalanced. The classifier can simply label the samples as the majority class to achieve good accuracy." We now have clarified in the manuscript that the data of the three classes are balanced.

Extended SSL vs SL experiment of lung and lymph node cancer subsection in Results

Because the number of classes in lung images is three and the number of images in each class is balanced (5,000 per class), the accuracy is used for the evaluation.

The number of three classes were listed in supplementary Table 5.

Supplementary Table 5. Training and testing set for lung models

Model	Class	Dataset-Lung		
		Training set		Testing set
		Labeled	Unlabeled	
Lung-5%-SSL	adenocarcinoma	250	3750	1,000
	squamous cell carcinoma	250	3750	1,000
	benign	250	3750	1,000
	Total	750/5%	11,250/75%	3,000/20%
Lung-20%-SSL	adenocarcinoma	1,000	3,000	1,000
	squamous cell carcinoma	1,000	3,000	1,000
	benign	1,000	3,000	1,000
	Total	3,000/20%	9,000/60%	3,000/20%
Lung-5%-SL	adenocarcinoma	250	-	1,000
	squamous cell carcinoma	250	-	1,000
	benign	250	-	1,000
	Total	750/5%	-	3,000/20%
Lung-20%-SL	adenocarcinoma	1,000	-	1,000
	squamous cell carcinoma	1,000	-	1,000
	benign	1,000	-	1,000
	Total	3,000/20%	-	3,000/20%
Lung-80%-SL	adenocarcinoma	4,000	-	1,000
	squamous cell carcinoma	4,000	-	1,000
	benign	4,000	-	1,000
	Total	12,000/80%	-	3,000/20%

R3-Q2 :

There are also some recent works [1,2] on the annotation strategy for WSI, where only a fraction of positive samples are annotated, and the other samples are unlabeled. This strategy may further reduce the annotation effort, as the experts do not need to carefully ensure that there are no positive instances in the negative image. Then, network training can be performed with loss calibration or positive-unlabeled learning [3]. Although these works are originally developed for cell detection, they can be adapted for WSI classification. The authors could consider discussing the possibility of such annotation in future work.

References

- [1] Li et al., A Novel Loss Calibration Strategy for Object Detection Networks Training on Sparsely Annotated Pathological Datasets, MICCAI 2020.
- [2] Zhao et al., Positive-unlabeled Learning for Cell Detection in Histopathology Images with Incomplete Annotations, arXiv:2106.15918
- [3] Fabien et al., Learning from positive and unlabeled examples, Algorithmic Learning Theory, 2000.

Thank you very much for your suggestions. How to use a small number of labeled samples to achieve better results in machine learning has always been a hot topic of research in this field. The recent work you recommended [1,2] made new attempts and are really insightful to us. We will refer to these works for future exploration of their methods to pathological image recognition problems.

We have discussed the recent studies in the last paragraph of Discuss section of revised manuscript.

SSL may have excellent potentials to overcome the bottleneck of insufficient labeled data as in many medical domains. In addition, we have noticed some other recent works [51, 52], which have made new strategy on the sparse and incomplete annotations to reduce the annotation effort for cell detection. This strategy is also applicable to annotations in our WSIs, and the unlabeled data is useful for SSL. In future work, how to make annotations and use unlabeled data more effectively should be further studied to improve the efficiency of medical AI development.

Reviewers' Comments:

Reviewer #2:

Remarks to the Author:

All comments were addressed and I have no additional questions.